# AMLRIS: Alignment-aware Masked Learning for Referring Image Segmentation

**Tongfei Chen**[1][*] **Shuo Yang**[3][*] **Yuguang Yang**[3,2][*] **Linlin Yang**[4][†] **Runtang Guo**[7],
**Changbai Li**[5], **He Long**[1], **Chunyu Xie**[6,2][†] **Dawei Leng**[2][‡] **Baochang Zhang**[1][‡]

[1]School of Artificial Intelligence, Beihang University, China
[2]360 AI Research, Qihoo 360, China
[3]School of Electronic Information Engineering, Beihang University, China
[4]State Key Laboratory of Media Convergence and Communication, China
  Communication University of China, China
[5]Hangzhou International Innovation Institute, Beihang University, China
[6]Institute of Unmanned System, Beihang University, China
[7]School of Automation Science and Electrical Engineering,   Beihang University, China

## Abstract

Referring Image Segmentation (RIS) aims to segment the object in an image uniquely referred to by a natural language expression. However, RIS training often contains hard-to-align and instance-specific visual signals; optimizing on such pixels injects misleading gradients and drives the model in the wrong direction. By explicitly estimating pixel-level vision–language alignment, the learner can suppress low-alignment regions, concentrate on reliable cues, and acquire more generalizable alignment features. In this paper, we propose Alignment-Aware Masked Learning (AML), a simple yet effective training strategy that quantifies region–referent alignment (PMME) and filters out unreliable pixels during optimization (AFM). Specifically, each sample first computes a similarity map between visual and textual features, and then masks out pixels falling below an adaptive similarity threshold, thereby excluding poorly aligned regions from the training process. AML does not require architectural changes and incurs no inference overhead, directing attention to the areas aligned with the textual description. Experiments on the RefCOCO (vanilla/+/g) datasets show that AML achieves state-of-the-art results across all 8 splits, and beyond improving RIS performance, AML also enhances the model's robustness to diverse descriptions and scenarios. Code is available at https://github.com/pipashu1/AMLRIS.

## 1 Introduction

Referring Image Segmentation (RIS) (Hu et al., 2016; Cheng et al., 2014) aims to segment the object in an image that is uniquely identified by a natural language expression. Unlike conventional segmentation tasks, RIS requires precise cross-modal reasoning under sparse supervision—each training sample typically includes only one annotated object. However, resolving such expressions often depends on understanding the visual context surrounding the referent, including other objects, spatial relations, and appearance-based contrasts. For instance, identifying "the giraffe closest to people" (Fig. 1a) requires distinguishing multiple similar objects based on proximity, while resolving "lower broccoli" (Fig. 1b) requires fine-grained attention to localized attributes and utilize this information to determine the full target entity. This contextual grounding is crucial, yet difficult to learn from limited pixel-level supervision.

---

[*]Equal contribution.{tfchen, shuo1yang, guangbuaa}@buaa.edu.cn
[†]Corresponding author.`lyang@cuc.edu.cn`, `xiechunyu@360.cn`
[‡]Project leader.

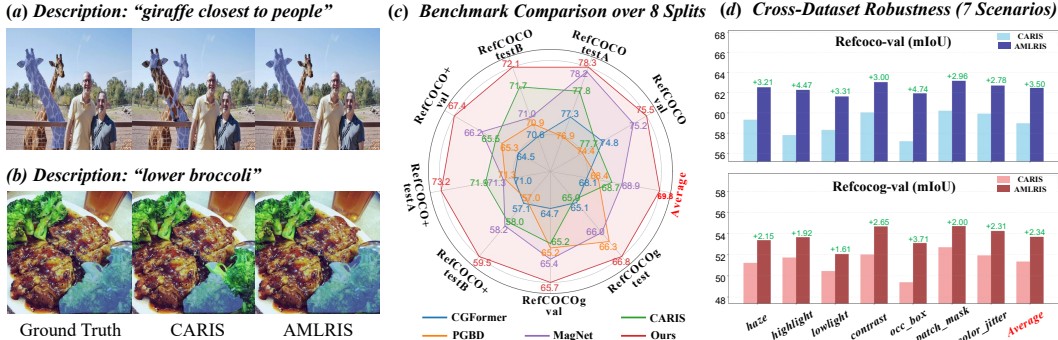

Figure 1: Overall results of AMLRIS, including qualitative examples, benchmark comparisons (oIoU), and cross-dataset robustness evaluation, where the model is trained only on RefCOCO+ and evaluated on RefCOCOg and RefCOCO under seven perturbation scenarios.

To address the supervision bottleneck, prior works (Wang et al., 2022a; Yang et al., 2022; Zhang et al., 2022; Ding et al., 2022; Liu et al., 2023a; Zhao et al., 2023; Xie et al., 2023; Zhou et al., 2024; Yang et al., 2025; Xie et al., 2025) have introduced architectural modules to improve cross-modal alignment. LAVT (Yang et al., 2022) incorporates pixel–word attention in the encoder to fuse visual and linguistic cues early. CARIS (Liu et al., 2023c) enhances decoding with bidirectional cross-attention and contextual prompts, while DETRIS (Huang et al., 2025) promotes hierarchical alignment through densely connected language adapters, allowing language signals to influence visual representations across multiple semantic levels. These designs aim to compensate for weak supervision by strengthening vision–language interaction. **However, without reliable supervision beyond the referred object, modules trained under dense loss may overfit to unrelated regions, allowing misaligned gradients to dominate training.**

To address the above limitations, we propose Alignment-Aware Masked Learning (AML), a simple yet effective training strategy that improves RIS by selectively filtering out unreliable pixels during optimization. **The core idea is to decouple learning from poorly aligned regions, allowing the model to concentrate gradient updates on more trustworthy visual–textual correspondences.** Therefore, in the first forward, we introduce a PatchMax Matching Evaluation (PMME) to compute a fine-grained similarity map, where each visual patch is matched to its most similar language token to quantify the patch-text alignment. Note that in many RIS architectures, the vision and language backbones are not jointly pretrained and often output features with mismatched dimensionalities. To enable meaningful similarity computation, we introduce a Johnson–Lindenstrauss random projection that maps both modalities into a common embedding space. This projection preserves pairwise distances and angular structures with high probability, ensuring that the cross-modal geometry remains intact while aligning feature dimensions (we have provided a thorough theorem for this in Sec. 3.2).

Subsequently, based on the similarity map, we construct an Alignment-Aware Filtering Mask (AFM), masking out pixels whose similarity falls below a pre-set threshold. The model is then optimized on the remaining well-aligned regions in the second forward. AML improves optimization stability by filtering out weak learning signals early in training. As noted by Zheng et al. (2023), this reduces the search space and prevents overfitting to misaligned regions. **In RIS, where fine-grained alignment is key to distinguishing similar objects, AML guides attention toward the most relevant regions—for example, focusing on discriminating the correct individual in expressions like "giraffe closest to people" (Fig. 1a).**

Extensive experiments on RefCOCO, RefCOCO+, and RefCOCOg demonstrate that AML consistently improves performance across multiple backbones and achieves state-of-the-art results. Moreover, we show that AML enhances model robustness in occluded or noisy conditions, confirming the value of alignment-aware supervision under limited annotations. Our contributions can be summarized as follows:

- We propose an Alignment-Aware Masked Learning (AML) framework, which selectively filters out poorly aligned pixels based on a patch-level cross-modal similarity map.

- We introduce PatchMax Matching Evaluation (PMME) to quantify cross-modal feature alignment and Alignment-aware Filtering Masking (AFM) to enable fine-grained region selection.

- Extensive experiments on RefCOCO, RefCOCO+, and RefCOCOg demonstrate that AMLRIS achieves state-of-the-art performance across all 8 splits with consistent improvements over previous SOTA methods (Fig. 1c), and exhibits superior cross-dataset robustness, surpassing the baseline under diverse perturbation scenarios (Fig. 1d).

## 2 RELATED WORK

**Referring Image Segmentation.** Referring Image Segmentation (RIS) aims to segment a target object in an image based on a referring natural language expression. Most methods (Wang & Ye, 2024; Wu et al., 2025; Shi et al., 2018; Chen et al., 2019) follow a two-stage paradigm: visual and linguistic features are first extracted separately and then fused for segmentation. Depending on the fusion stage, RIS methods can be categorized into late fusion and early fusion approaches. **Late fusion** methods perform cross-modal alignment after feature extraction, typically within the decoder. VLT (Ding et al., 2022), CARIS (Liu et al., 2023c), and ReSTR (Kim et al., 2022) adopt cross-attention to connect visual patches and language tokens, while CRIS (Wang et al., 2022b) leverages CLIP for pixel-level adaptation. However, these methods often suffer from limited vision-language interaction and fail to establish trustworthy image region-context correspondences. **Early fusion** methods inject language features into the visual encoder to enable language-aware representation during encoding. LAVT (Yang et al., 2022) introduces pixel–word attention, CGFormer (Tang et al., 2023) uses gated cross-modal fusion, and DETRIS (Huang et al., 2025) distributes lightweight language adapters throughout the visual backbone. These designs (Li et al., 2025; Yang et al., 2022; Huang et al., 2025; Ding et al., 2022; Liu et al., 2023c; Kim et al., 2022; Wang et al., 2022b; Yang et al., 2022; Tang et al., 2023; Huang et al., 2025; Long et al., 2025; Zhou et al., 2025; Yang et al., 2024a) strengthen vision–language coupling and enhance referential grounding, especially in complex scenes.

However, existing methods attempt to model all spatial and semantic relationships through increasingly intricate fusion mechanisms, implicitly assuming that all regions are equally informative. In contrast, our approach takes the opposite view: rather than modeling every relation, we first eliminate regions that are poorly aligned with the expression. By masking out these ambiguous areas, AML allows the model to focus more on trustworthy regions and learn referential cues from clearer, more consistent visual–textual correspondences.

**Masking-based Data Augmentation.** More recently, data-centric approaches (Cubuk et al., 2018; 2020; Touvron et al., 2021; Lee et al., 2024; Chng et al., 2024; Ha et al., 2024) have emerged to supplement the limited training signal. However, designing effective data augmentation for RIS poses unique challenges. Standard transformations used in vision tasks (Shorten & Khoshgoftaar, 2019; Alomar et al., 2023; Perez & Wang, 2017; Yu et al., 2024; Lyu et al., 2024), such as horizontal flipping or color jittering, can easily violate the referential integrity of the expression. For instance, flipping the image invalidates spatial expressions like "on the left", while color jitter may distort expressions such as "the woman in red". Unlike classification, RIS requires that any image transformation preserve the semantic link between expression and referent. To address this, MaskRIS (Lee et al., 2024) introduces distortion-aware augmentation strategies that avoid operations likely to break semantic consistency. Other efforts (Chng et al., 2024) leverage masking-based training signals: Mask Grounding applies token-level masking on the text side and requires the model to recover masked words using visual cues, thereby enhancing alignment between image regions and key linguistic units. NeMo (Ha et al., 2024) adopts a vision-side strategy, embedding hard negative distractors into mosaic images to increase discrimination difficulty and force the model to resolve fine-grained ambiguity.

Although these methods enhance supervision, they still apply full-pixel loss, allowing gradients from irrelevant or misaligned regions to dominate training. In contrast, AML directly suppresses noisy gradients by masking out poorly aligned pixels based on patch–token similarity, guiding learning toward reliable regions and improving referent grounding under sparse supervision.

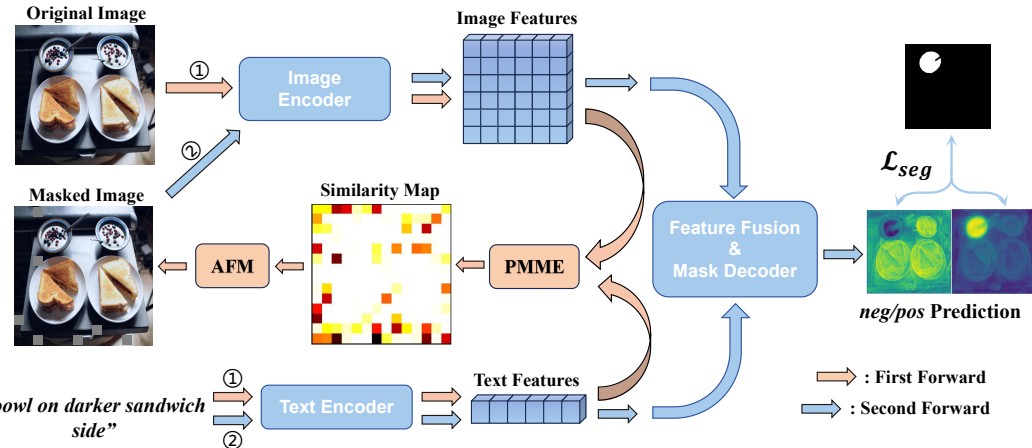

Figure 2: Overview of Alignment-aware Masked Learning (AML) framework.

## 3 METHOD

### 3.1 PRELIMINARY

**RIS Formulation.** Given an image $I \in \mathbb{R}^{H \times W \times 3}$ and a referring expression $T = \{w_1, w_2, \ldots, w_L\}$ of length $L$, the goal of RIS is to predict a binary mask $M \in \{0,1\}^{H \times W}$, where each pixel indicates whether it belongs to the referred object.

**Baseline Model.** We build upon CARIS (Liu et al., 2023c), a recent context-aware baseline for RIS. CARIS adopts a dual-branch architecture with a Swin-B image encoder $\phi_v$ and a BERT-based text encoder $\phi_t$ to extract multi-scale visual features $\{V_{enc}^k\}_{k=1}^K$ at $K$ scales and token embeddings with $D_t$-dimensional $T_{enc} \in \mathbb{R}^{N_l \times D_t}$, respectively. Then, a mask decoder is used to produce the positive region prediction $M_{pos}$ and negative region prediction $M_{neg}$. And the probabilities of the position $(i,j)$ for the referred object is:

$$P^{(i,j)} = \frac{\exp(M_{\text{pos}}^{(i,j)})}{\exp(M_{\text{pos}}^{(i,j)}) + \exp(M_{\text{neg}}^{(i,j)})}. \tag{1}$$

The loss function is calculated as follows:

$$\mathcal{L}_{\text{seg}} = -\frac{1}{HW} \sum_{i=1}^{H} \sum_{j=1}^{W} \left[ y^{(i,j)} \log P^{(i,j)} + (1 - y^{(i,j)}) \log(1 - P^{(i,j)}) \right], \tag{2}$$

where $y^{(i,j)} = 1$ for positive pixels and $y^{(i,j)} = 0$ for negative pixels.

### 3.2 PATCHMAX MATCHING EVALUATION

**PMME with Random Projection Alignment.** To diagnose the modality gap between vision and language representations in referring image segmentation (RIS), we introduce PatchMax Matching Evaluation (PMME), which quantifies the semantic alignment between visual patches and language tokens.

A key challenge lies in the dimensional mismatch between visual and language features, which makes direct similarity computation infeasible. To address this, we project both modalities into a common embedding space via modality-specific random linear mappings (Achlioptas, 2003). Such projections approximately preserve pairwise distances and inner products with high probability, allowing reliable similarity computation in the embedding space (see Theorem1). Specifically, given the deepest layer of visual features $V_{enc}^K = \{v_m\}_{m=1}^{H_f W_f} \in \mathbb{R}^{H_f W_f \times D_i}$ and textual features $T_{enc} = \{u_n\}_{n=1}^{N_l} \in \mathbb{R}^{N_l \times D_t}$, we perform a flattening operation followed by an $\ell_2$-normalization process:

$$\tilde{v}_m = v_m / \|v_m\|_2, \quad \tilde{u}_n = u_n / \|u_n\|_2, \tag{3}$$

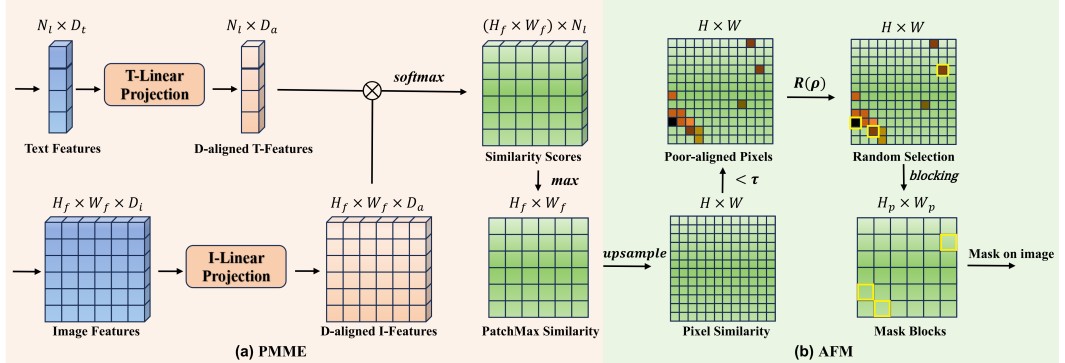

Figure 3: Architecture for PMME and AFM modules.

and thus we have the normalized $\tilde{V} = \{\tilde{v}_m\}_{m=1}^{H_f W_f}$ and $\tilde{T} = \{\tilde{u}_n\}_{n=1}^{N_l}$. Then we project them into a $D_a$-dimensional space using *i.i.d.* gaussian matrices $W_i \in \mathbb{R}^{D_i \times D_a}$ and $W_t \in \mathbb{R}^{D_t \times D_a}$:

$$V' = \tilde{V} W_i^\top \quad T' = \tilde{T} W_t^\top, \tag{4}$$

$$[W_i]_{pq} \sim \mathcal{N}(0, 1/D_a) \quad [W_t]_{pq} \sim \mathcal{N}(0, 1/D_a), \tag{5}$$

where $[W_i]_{pq}$, $[W_t]_{pq} \in \mathbb{R}$ are the elements for $W_i$ and $W_t$, respectively. Then we perform dot product and SoftMax for our raw alignment matrix:

$$S_{\text{norm}} = \text{SoftMax}(V' T'^\top) \in \mathbb{R}^{(H_f \times W_f) \times N_l}, \tag{6}$$

where SoftMax is applied across each row. This yields a probability distribution over all language tokens for each visual patch, quantifying their alignment confidence. To derive a fine-grained patch-level alignment signal, we select for each visual patch its most strongly aligned token:

$$S^{(i,j)} = \max_{1 \le k \le N_l} S_{\text{norm}}^{(i,j,k)} \in \mathbb{R}^{(H_f \times W_f)}. \tag{7}$$

where $S^{(i,j)} \in [0, 1]$ indicates the peak alignment confidence of patch $(i, j)$. The resulting metric $S \in \mathbb{R}^{H_f \times W_f}$ serves as a fine-grained alignment heatmap, which is later used to filter unreliable regions during training. Furthermore, we rigorously demonstrate the validity of this computation.

**Theorem1 Cross-Modal Inner Product Preservation via Gaussian Random Mapping.** Let $\mathcal{Z} = \left\{ z_{mn} = [\tilde{v}_m, \tilde{u}_n]^T \in \mathbb{R}^{D_i + D_t} \right\}$ be the joint embedding space. A block-diagonal projection can be defined as:

$$\widetilde{W} = \frac{1}{\sqrt{2}} \operatorname{diag}(W_i, W_t) \in \mathbb{R}^{(D_i + D_t) \times 2D_a}. \tag{8}$$

Then, for any distortion $\varepsilon \in (0, 1)$ and failure probability $\sigma \in (0, 1)$, if:

$$D_a \ge 8 \log(H_f W_f N_l / \sigma) / \varepsilon^2, \tag{9}$$

then with probability at least $1 - \sigma$, the projection $\widetilde{W}$ preserves all pairwise distances in $\mathcal{Z}$:

$$|\langle W_i \tilde{v}_m, W_t \tilde{u}_n \rangle - \langle \tilde{v}_m, \tilde{u}_n \rangle| \le \varepsilon, \quad \forall m, n. \tag{10}$$

This theorem ensures that all pairwise dot products between the joint embeddings $(v_m, u_n)$ are approximately preserved under the random block-diagonal projection $\widetilde{W}$. The guarantee holds uniformly with high probability as long as the projection dimension $D_a$ satisfies Eq. 9. The thorough proof can be found in **Appendix F**.

### 3.3 ALIGNMENT-AWARE FILTERING MASKING (AFM)

To assess fine-grained visual-language alignment, we first upsample the patch-level similarity matrix $S \in \mathbb{R}^{H_f \times W_f}$ to the original image resolution using bilinear interpolation, yielding a dense pixel-wise similarity map $S_{\text{pixel}} \in \mathbb{R}^{H \times W}$:

$$S_{\text{pixel}} = \text{BilinearUpsample}(S). \tag{11}$$

This upsampling step enhances spatial consistency by propagating alignment scores to neighboring pixels. We then identify weakly aligned pixels—those with similarity below a threshold $\tau$—and collect their positions into a candidate set:

$$\mathcal{P}_{\text{weak}} = \{(m, n) \in [1, H] \times [1, W] \mid S_{\text{pixel}}^{(m,n)} < \tau\}. \tag{12}$$

To prevent over-filtering and encourage generalization, we randomly retain a proportion $1 - \rho$ of these weak pixels:

$$\mathcal{P}_{\text{selected}} = \text{DropOut}\left(\mathcal{P}_{\text{weak}}, \rho\right). \tag{13}$$

Next, we aggregate the selected pixel-level mask into patch-level binary blocks. The image is partitioned into non-overlapping $B^h \times B^w$ blocks, where $B^h$ and $B^w$ refer to the height and width of each image patch, each corresponding to a downsampled mask element:

$$M_{\text{block}}^{(p,q)} = \max_{(m,n) \in \mathcal{B}^{(p,q)}} \mathbb{I}[(m, n) \in \mathcal{P}_{\text{selected}}], \tag{14}$$

where $\mathcal{B}^{(p,q)} = [pB^h, (p + 1)B^h] \times [qB^w, (q + 1)B^w]$ denotes the pixel range of the $(p, q)$-th block, and $\mathbb{I}[\cdot]$ is the indicator function. The max operator ensures that a block is masked if any of its constituent pixels are poorly aligned, following a conservative "any-triggers-all" policy to suppress noisy gradients.

Finally, we apply this binary mask to the input image, zeroing out the selected blocks:

$$\tilde{I}(\mathcal{B}^{(p,q)}) = I(\mathcal{B}^{(p,q)}) \odot (1 - M_{\text{block}}^{(p,q)}), \tag{15}$$

where $\odot$ denotes element-wise multiplication. This masking suppresses gradients from unreliable regions, allowing the model to focus on well-aligned areas during training, ultimately improving the overall alignment quality (see **Appendix E**).

## 3.4 AML Training Framework

To integrate alignment-aware augmentation while maintaining compatibility with the baseline optimization, as shown in Fig. 2, we adopt a two-stage training scheme with shared model parameters across stages.

**In the first stage**, the original image $I$ and referring expression $T$ are passed through the vision and language encoders to compute the similarity map $S$ and derive the binary mask $M_{\text{block}}$ via our alignment-aware filtering. The masked image $\tilde{I}$ is constructed by zeroing out poorly aligned regions. This stage is forward-only, incurs no gradient computation, and adds slight overhead (a 4.9% memory and 17.2% training time overhead compared with CARIS, detailed in the **Appendix G.2**).

**In the second stage**, $\tilde{I}$ and $T$ are fed into the baseline model to perform standard segmentation and compute loss. **Only this stage updates the model parameters.** Since both stages share weights and the masking stage is lightweight, the total training cost remains comparable to the baseline.

This framework enables the model to benefit from alignment-informed masking while preserving training efficiency and objective consistency. Notably, **during inference, the masking stage is skipped, and the model operates on the original input.** Despite never being trained on unmasked images, the model exhibits improved generalization and robustness, outperforming both the baseline and random masking variants (see Figure 4).

## 4 Experiment

### 4.1 Experimental Setup

**Benchmark & Metric.** We evaluate our method on three widely adopted Referring Image Segmentation (RIS) benchmarks: RefCOCO, RefCOCO+, and RefCOCOg. These datasets provide diverse challenges in terms of expression complexity and object density (See **Appendix A.1** for detailed introduction). And we adopt overall Intersection-over-Union (oIoU), mean Intersection-over-Union (mIoU), and Precision at IoU thresholds (P@$X$, $X \in \{0.5, 0.7, 0.9\}$) to evaluate the predicted mask (see **Appendix B**).

---

**Algorithm 1** Alignment-Aware Masked Learning (AML) for RIS

---

1: **Input:** training set $\{(I, T, Y)\}$; image encoder $\phi_v$; text encoder $\phi_t$; RIS model $f_\theta$; random Gaussian matrices $W_i \in \mathbb{R}^{D_i \times D_a}$, $W_t \in \mathbb{R}^{D_t \times D_a}$; threshold $\tau$; dropout ratio $\rho$; block size $(B^h, B^w)$.

2: **Output:** updated parameters $\theta$.

3: **for** each mini-batch $(I, T, Y)$ **do**

4:    */* Stage I: PMME-based alignment-aware masking (forward-only) */*

5:    $V_{\text{enc}}^K \leftarrow \phi_v(I)$   *// deepest visual feature, flattened to $\mathbb{R}^{H_f W_f \times D_i}$*

6:    $T_{\text{enc}} \leftarrow \phi_t(T)$   *// token features $\in \mathbb{R}^{N_l \times D_t}$*

7:    $\tilde{V} \leftarrow \ell_2\text{-normalize}(V_{\text{enc}}^K)$,   $\tilde{T} \leftarrow \ell_2\text{-normalize}(T_{\text{enc}})$

8:    $V' \leftarrow \tilde{V} W_i^\top$,   $T' \leftarrow \tilde{T} W_t^\top$   *// project to $D_a$-dim space*

9:    $S_{\text{norm}} \leftarrow \text{SoftMax}(V' T'^\top)$   *// row-wise over tokens*

10:    For each patch $(i, j)$, compute $S^{(i,j)} = \max_{1 \le k \le N_l} S_{\text{norm}}^{(i,j,k)}$

11:    $S_{\text{pixel}} \leftarrow \text{BilinearUpsample}(S)$   *// $S_{\text{pixel}} \in \mathbb{R}^{H \times W}$*

12:    $\mathcal{P}_{\text{weak}} \leftarrow \{(m, n) \mid S_{\text{pixel}}^{(m,n)} < \tau\}$

13:    $\mathcal{P}_{\text{selected}} \leftarrow \text{DropOut}(\mathcal{P}_{\text{weak}}, \rho)$

14:    Initialize block mask $M_{\text{block}}^{(p,q)} \leftarrow 0$ for all blocks.

15:    Partition the image into non-overlapping blocks of size $B^h \times B^w$.

16:    **for** each block $(p, q)$ **do**

17:       Define $\mathcal{B}^{(p,q)} = [pB^h, (p+1)B^h) \times [qB^w, (q+1)B^w)$.

18:       **if** there exists $(m, n) \in \mathcal{B}^{(p,q)}$ such that $(m, n) \in \mathcal{P}_{\text{selected}}$ **then**

19:          $M_{\text{block}}^{(p,q)} \leftarrow 1$

20:       **end if**

21:    **end for**

22:    $\tilde{I} \leftarrow I$

23:    **for** each block $(p, q)$ **do**

24:       **if** $M_{\text{block}}^{(p,q)} = 1$ **then**

25:          $\tilde{I}(\mathcal{B}^{(p,q)}) \leftarrow 0$   *// zero out poorly aligned blocks*

26:       **end if**

27:    **end for**

28:    */* Stage II: RIS training on masked image */*

29:    $(M_{\text{pos}}, M_{\text{neg}}) \leftarrow f_\theta(\tilde{I}, T)$

30:    Compute $P^{(i,j)}$ as in Eq. (1).

31:    Compute $\mathcal{L}_{\text{seg}}$ as in Eq. (2) using $Y$.

32:    Update $\theta \leftarrow \theta - \eta \nabla_\theta \mathcal{L}_{\text{seg}}$.

33: **end for**

---

**Training Details.** We integrate AML into several representative RIS frameworks, including DE-TRIS (Huang et al., 2025), CARIS (Liu et al., 2023c) and ReLA (Liu et al., 2023a). Without special notation, the baseline model is CARIS. We set $\tau = 0.4$, $\rho = 0.25$, $H_P \times W_p = 32 \times 32$, $D_a = 2048$. More training details in **Appendix C**.

## 4.2 COMPARATIVE EXPERIMENT ON RIS DATASETS

**CARIS with AML achieves new SOTA performance on the RIS task.** As shown in Table 1, our method consistently outperforms existing approaches across RefCOCO, RefCOCO+, RefCOCOg, and the combined setting, in terms of both mIoU and oIoU. On RefCOCO, CARIS+AML improves over the baseline by +1.12%, +0.50%, and +0.43% mIoU, and +0.80%, +0.50%, and +0.42% oIoU on val/testA/testB, respectively. On RefCOCO+, we observe consistent gains: +2.00%, +1.10%, and +1.92% mIoU, and +1.83%, +1.33%, and +1.54% oIoU. On RefCOCOg, CARIS+AML surpasses the baseline by +1.22% (test) in mIoU, and +1.78% in oIoU. These consistent improvements validate the effectiveness of AML in enhancing semantic precision and spatial consistency, especially under complex linguistic expressions. Moreover, under the combined training setting, our approach further improves upon the baseline by +0.53% (RefCOCO val), +0.96% (RefCOCO+ val), and +1.40% (RefCOCOg val) in oIoU, while maintaining superior performance across all test splits.

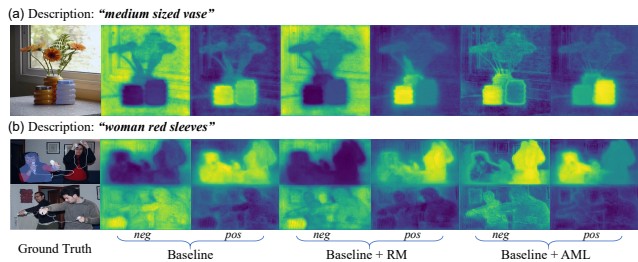
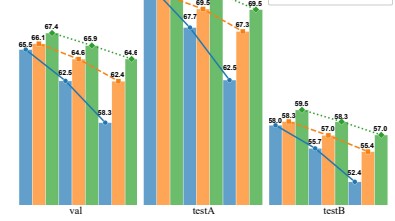

(a) Comparison of prediction maps: Baseline, Random Mask (RM), and AML. Brighter regions indicate higher scores. A location is predicted as target when $pos > neg$.

(b) Comparative performances with different mask strategies using occluded images on RefCOCO+.

Figure 4: Qualitative and quantitative comparison with the Random Mask strategy.

Table 1: Comparison with SOTA methods. * denotes the reproduced results across all experiments. **Bold** and underline numbers indicate the best and the second best performance. The last column is the average score across all available splits. Gray texts denote large language model based methods.

| Method | RefCOCO | | | RefCOCO+ | | | RefCOCOg | | Avg |
|---|---|---|---|---|---|---|---|---|---|
| | val | testA | testB | val | testA | testB | val | test | |
| *Standard: Training on the training split of each dataset.* | | | | | | | | | |
| **mIoU** | | | | | | | | | |
| CRIS (Wang et al., 2022a) | 70.47 | 73.18 | 66.10 | 62.27 | 68.08 | 53.68 | 59.87 | 60.36 | 64.3 |
| ETRIS (Xu et al., 2023) | 71.06 | 74.11 | 66.66 | 62.23 | 68.51 | 52.79 | 60.28 | 60.42 | 64.5 |
| BarLeRIa (Wang et al., 2024) | 72.40 | 75.90 | 68.30 | 65.00 | 70.80 | 56.90 | - | - | - |
| LAVT (Yang et al., 2022) | 74.46 | 76.89 | 70.94 | 65.81 | 70.97 | 59.23 | 63.34 | 63.62 | 68.0 |
| DETRIS* (Huang et al., 2025) | 75.64 | 77.39 | 73.11 | 68.47 | 73.03 | 60.00 | - | - | - |
| CGFormer (Tang et al., 2023) | 76.93 | 78.70 | 73.32 | 68.56 | 73.76 | 61.72 | 67.57 | 67.83 | 71.1 |
| PGBD (Wu et al., 2025) | 76.45 | 78.57 | 73.14 | 68.82 | 73.84 | 61.56 | 68.42 | 67.99 | 71.1 |
| CARIS* (Liu et al., 2023c) | 76.77 | 79.03 | 74.56 | 69.33 | 74.51 | 62.69 | 68.87 | 68.51 | 71.8 |
| MagNet (Chng et al., 2024) | 77.43 | 79.43 | 74.11 | 70.10 | 74.50 | 63.59 | 68.53 | 69.15 | 72.1 |
| Ours | **77.89** | **79.53** | **74.99** | **71.33** | **75.61** | **64.61** | **69.24** | **69.73** | **72.9** |
| **oIoU** | | | | | | | | | |
| LAVT (Yang et al., 2022) | 72.73 | 75.82 | 68.79 | 62.14 | 68.38 | 55.10 | 61.24 | 62.09 | 65.8 |
| DETRIS* (Huang et al., 2025) | 74.05 | 76.68 | 71.33 | 65.39 | 70.82 | 55.82 | - | - | - |
| NeMo (Ha et al., 2024) | 74.48 | 76.32 | 71.51 | 62.86 | 69.92 | 55.56 | 64.40 | 64.80 | 67.5 |
| ReMamber (Yang et al., 2024b) | 74.54 | 76.74 | 70.89 | 65.00 | 70.78 | 57.53 | 63.90 | 64.00 | 67.9 |
| CGFormer (Tang et al., 2023) | 74.75 | 77.30 | 70.64 | 64.54 | 71.00 | 57.14 | 64.68 | 65.09 | 68.1 |
| PGBD (Wu et al., 2025) | 74.43 | 76.89 | 70.91 | 65.27 | 71.26 | 56.98 | 65.21 | 66.31 | 68.4 |
| CARIS* (Liu et al., 2023c) | 74.65 | 77.83 | 71.70 | 65.54 | 71.86 | 57.97 | 65.15 | 65.00 | 68.7 |
| MagNet (Chng et al., 2024) | 75.24 | 78.24 | 71.05 | 66.16 | 71.32 | 58.14 | 65.36 | 66.03 | 68.9 |
| Ours | **75.45** | **78.33** | **72.12** | **67.37** | **73.19** | **59.51** | **65.67** | **66.78** | **69.8** |
| *Combined: Training on the combination of three datasets.* | | | | | | | | | |
| **oIoU** | | | | | | | | | |
| PixelLM-7B (Ren et al., 2024) | 73.0 | 76.5 | 68.2 | 66.3 | 71.7 | 58.3 | 69.3 | 70.5 | 69.2 |
| GSVA-7B (Xia et al., 2024) | 76.4 | 77.4 | 72.8 | 64.5 | 67.7 | 58.6 | 71.1 | 72.0 | 70.1 |
| PerceptionGPT-7B (Ren et al., 2024) | 75.1 | 78.6 | 71.7 | 68.5 | 73.9 | 61.3 | 70.3 | 71.7 | 71.4 |
| PolyFormer (Liu et al., 2023b) | 74.82 | 76.64 | 71.06 | 67.64 | 72.89 | 59.33 | 67.76 | 69.05 | 69.2 |
| CARIS* (Liu et al., 2023c) | 76.67 | 78.24 | 71.32 | 67.64 | 72.77 | 60.87 | 67.44 | 69.17 | 70.5 |
| MagNet (Chng et al., 2024) | 76.55 | 78.27 | 72.15 | 68.10 | 73.64 | **61.81** | 67.79 | 69.29 | 70.9 |
| Ours | **77.20** | **79.53** | **73.61** | **68.60** | **73.76** | 61.55 | **68.84** | **70.01** | **71.9** |

**Effectiveness Across Multiple Baselines.** To demonstrate the generality and early-stage efficiency (effectiveness in the initial few epochs) of AML, we apply it to multiple representative RIS baselines. As shown in Table 2, AML consistently improves oIoU across all RefCOCO splits, boosting CARIS by +1.83%, +1.33%, +1.54%, and DETRIS by +0.80%, +0.38%, +0.98% on val, testA, and testB, respectively.

**Effectiveness in the Early Training Stage.** In multi-object grounding experiments on the GRES dataset, integrating AML into ReLA yields consistent early-stage benefits (at 4999 iterations, see **Appendix G.3**), AML+ReLA reaches 30.86 cIoU and 21.08 gIoU, compared to 28.49 and 17.33 for vanilla ReLA (Table 3). Similarly, under the same ten-epoch protocol as MagNet (Chng et al.,

Table 2: AML with more baseline models on Re-fCOCO+ (oIoU).

| Method | Image Encoder | Text Encoder | val | testA | testB |
|--------|---------------|--------------|-----|-------|-------|
| *Fully-supervised (oIoU)* | | | | | |
| CARIS | Swin-B | BERT | 65.54 | 71.86 | 57.97 |
| +AML | Swin-B | BERT | **67.37** | **73.19** | **59.51** |
| *Parameter-Efficient (oIoU)* | | | | | |
| DETRIS | DINOv2-B | CLIP | 65.39 | 70.82 | 55.82 |
| +AML | DINOv2-B | CLIP | **66.20** | **71.20** | **56.80** |

Table 3: Early-stage comparisons across datasets and metrics.

| Method | Dataset (Metric) | |
|--------|------------------|------------------|
| *Single-object* | RefCOCO (oIoU) | RefCOCO+ (oIoU) |
| MagNet | 68.52 | 57.26 |
| **Ours** | **71.83 (+3.31)** | **63.79 (+5.53)** |
| *Multi-object* | GRES (cIoU) | GRES (gIoU) |
| ReLA | 28.49 | 17.33 |
| **Ours** | **30.86 (+2.37)** | **21.08 (+3.75)** |

Table 4: Threshold ($\tau$) ablation on val splits.

| $\tau$ | RefCOCO | RefCOCO+ | RefCOCOg |
|--------|---------|----------|----------|
| 0 (CARIS) | 74.65 | 65.54 | 65.15 |
| 0.2 | 75.00 | 66.99 | **65.78** |
| 0.3 | **75.81** | 66.82 | 65.13 |
| **0.4** | 75.45 | **67.37** | 65.67 |

Table 6: PMME phase ablation on RefCOCO+.

| Methods | P@0.5 | P@0.7 | P@0.9 | oIoU |
|---------|-------|-------|-------|------|
| CARIS | 78.85 | 71.76 | 33.48 | 65.54 |
| + Late PMME | 76.98 | 69.67 | 33.66 | 63.35 |
| + Pred PMME | 79.57 | 73.02 | 35.40 | 66.41 |
| **+ Early PMME** | **80.82** | **73.85** | **36.35** | **67.37** |

Table 5: Projection dimension ($D_a$) ablation.

| Dim ($D_a$) | oIoU | mIoU | Pr@50 | Pr@90 | JL Error ($\epsilon$) |
|-------------|------|------|-------|-------|-----------------------|
| 1024 | 66.82 | 70.83 | 80.15 | 36.27 | 0.2955 |
| **2048** | **67.37** | **71.33** | 80.82 | **36.35** | 0.2088 |
| 4096 | 67.08 | 71.21 | **80.85** | 36.22 | 0.1478 |

Table 7: PMME projection strategies (epoch 1).

| Method | P@0.5 | P@0.7 | P@0.9 | oIoU |
|--------|-------|-------|-------|------|
| CARIS | 8.66 | 2.70 | 0.25 | 20.24 |
| + Learnable | 10.36 | 6.02 | 0.92 | 16.92 |
| **+ Random** | **17.35** | **9.40** | **1.33** | **24.85** |

2024), our method(AML+CARIS) achieves 71.83% on RefCOCO and 63.79% on RefCOCO+, surpassing MagNet by +3.31% and +5.53% (Table 3). These improvements stem from discarding poorly aligned regions at the early training stage, which guides the optimization toward reliable correspondences (see **Appendix G.4**) and consistently boosts performance across both multi-object and single-object settings.

### 4.3 ROBUSTNESS UNDER CROSS-DATASET AND VISUAL PERTURBATIONS

**Setup.** To evaluate scene robustness, we train models on RefCOCO+ and test them on RefCOCO and RefCOCOg under multiple visual perturbations. The perturbation types cover illumination shifts, occlusion, and other scene-level variations, as detailed in **Appendix G.5**.

**Results.** As shown in Figure 1d, AML consistently improves resilience across settings. On Ref-COCO, the average mIoU increases from 58.98% to 62.48% **(+3.50)**. On RefCOCOg, the average mIoU rises from 51.36% to 53.69% **(+2.34)**. These results show that AML can handle varied textual descriptions across datasets while resisting diverse visual disturbances, leading to more reliable performance under real-world scene shifts.

### 4.4 ABLATION & ANALYSIS

**Ablation on $\tau$.** As shown in Table 4, fixing $\rho = 0.25$ (see **Appendix G.6**), small thresholds ($\tau \in [0.2, 0.4]$) yield consistent gains: on RefCOCO, mIoU increases from 74.65% ($\tau$=0) to 75.00% ($\tau$=0.2) and 75.45% ($\tau$=0.4); on RefCOCO+, from 65.54% to 66.99% and 67.37%. RefCOCOg also improves over baseline, peaking at 65.78% with $\tau$=0.2 (65.67% at $\tau$=0.4). These results suggest that small $\tau$ effectively filters poorly aligned regions, and that near-zero alignment can hinder optimization. Unless otherwise noted, we adopt $\tau$=0.4 as an average-best choice across datasets.

**Ablation on $D_a$.** As shown in Table 5, we compare $D_a$ on RefCOCO+: moving from $1024 \rightarrow 2048$ yields consistent gains (oIoU 66.82% $\rightarrow$ 67.37%, mIoU 70.83% $\rightarrow$ 71.33%, Pr@50 80.15% $\rightarrow$ 80.82%, Pr@90 36.27% $\rightarrow$ 36.35%). Pushing to 4096 brings only marginal changes. Meanwhile, the JL distortion bound decreases monotonically ($0.2955 \rightarrow 0.2088 \rightarrow 0.1478$), indicating better geometry preservation with larger $D_a$. Balancing accuracy and scalability, we adopt $D_a = 2048$

as the default, offering a favorable trade-off with a controlled bound ($\epsilon = 0.2088$) while nearing saturation on the main metrics.

**Ablation Study on Alignment Feature Stage in PMME.** We design three variants within the AML framework, as shown in Table 6: **Late PMME** (computes similarity after fusion operation, where features are highly entangled) performs the worst, likely due to the absence of semantic anchors when relying solely on entangled fused features. **Pred PMME** (uses the first-stage predicted mask as a coarse visual prior) ranks second, outperforming Late PMME in P@0.9 and oIoU, suggesting that even coarse localization guidance mitigates semantic misalignment. **Early PMME** (computes similarity using visual and textual features before cross-modal fusion) achieves the best performance across all metrics, improving P@0.9 and oIoU by +2.87% and +1.83% over the baseline, respectively. This validates computing alignment using clean, unimodal features before fusion preserves more accurate patch–token correspondences and enhances segmentation quality.

**Ablation Study on Projection Strategie in PMME.** We introduce a learnable projection variant of PMME for ablation, using trainable linear layers jointly optimized with the main network. Apart from the projection module, all other network components and training settings remain identical to ensure fair comparison. Implementation details are provided in Section 3.2. We assess the two strategies at the first training epoch to examine their influence on early-stage semantic alignment. As shown in Table 7, random projection outperforms both the CARIS baseline and the learnable variant by +7.9% and +4.6% in oIoU, respectively. This advantage is attributed to the stability of fixed mappings, which effectively suppress gradient noise and preserve cross-modal geometry, leading to faster and more reliable optimization.

**Robustness to Occluded Inputs.** To evaluate robustness under occlusion, we masked 0%, 30%, and 50% of image areas during inference. Compared to the control variant (CARIS+random mask), as shown in Figure 4b, AML consistently exhibits the smallest performance drops. On testA, AML reduced the oIoU drop from 9.4% to 3.7%, demonstrating superior robustness against degraded inputs compared to CARIS.

**Comparison of the Visualized Prediction Maps.** Following the robustness experimental setup with three compared methods, we visualize their negative and positive activation maps (bright regions indicate high response scores) in Figure 4a. For the same referring image, the baseline produces blurred prediction boundaries with activations leaking to semantically similar adjacent objects, while the random masking variant shows improved edge sharpness but still fails to precisely localize the correct mask position. In contrast, our method generates crisply defined boundaries and effectively filters out competing objects through negative activation suppression, demonstrating superior performance in both boundary precision and semantic discrimination. The observed characteristics indicate our method's superior semantic boundary modeling, which directly translates to more accurate referent localization and instance segmentation, as evidenced by the P@0.9 gains.

**Analysis of Training Overhead and Fairness.** Detailed in G.2, CARIS+AML increases training time by 17.2% per epoch but keeps total optimization steps identical. Under equal wall-clock time, CARIS+AML consistently outperforms CARIS (e.g., 66.7/70.8 vs. 65.5/69.3) and matches the 50-epoch CARIS performance in only 30 epochs, proving gains stem from alignment-aware masking rather than additional optimization.

# 5 CONCLUSION & LIMITATION

We propose Alignment-Aware Masked Learning (AML), a lightweight training strategy that improves referring image segmentation by filtering out poorly aligned pixels based on patch-level similarity. Extensive experiments on the RefCOCO family benchmarks demonstrate that AML consistently improves performance across different backbones and evaluation splits, without introducing any inference-time overhead or architectural modification. Its plug-and-play nature makes AML broadly applicable to existing RIS frameworks.

While AML proves broadly effective, it introduces a minor training overhead and shows sensitivity to a few hyperparameters. Future work will explore adaptive tuning and lighter variants to stabilize performance. We also plan to extend AML to broader tasks like video understanding and model families like alignment-pretrained foundation models.

ACKNOWLEDGEMENTS

This research was supported by the National Natural Science Foundation of China (Grant No. 62550184).

ETHICS STATEMENT

This submission adheres to the ICLR Code of Ethics and Code of Conduct. Our work does *not* involve human subjects, crowdsourced annotators, animal experiments, or sensitive personal data (e.g., face images linked to identity, medical or financial records); thus no IRB approval is required. All datasets used are publicly available for research under their respective licenses, which we follow without redistributing restricted content.

REPRODUCIBILITY STATEMENT

We provide complete algorithmic and architectural specifications, including loss definitions and hyperparameters, to ensure end-to-end reproducibility. After the review process, we will release the full source code, training/testing scripts, and model checkpoints to enable transparent verification.

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

# A  DATASET

## A.1  DETAILS

**RefCOCO** is a widely - adopted benchmark dataset sourced from the MSCOCO (Lin et al., 2014) dataset via a two - player game and contains 19,994 images annotated with 142,210 referring expressions for 50,000 objects. The dataset is partitioned into four subsets: 120,624 training samples, 10,834 validation samples, 5,657 samples for test A, and 5,095 samples for test B. The referring expressions have an average length of 3.6 words, and each image features at least two objects on average.

**RefCOCO+** is dataset consists of 141,564 referring expressions related to 49,856 objects in 19,992 images. The dataset also divided into four subsets: 120,624 training samples, 10,758 validation samples, 5,726 samples for test A, and 4,889 samples for test B. Distinctively, RefCOCO+ is designed to be more challenging than RefCOCO excluding certain types of absolute location words in the referring expressions.

**RefCOCOg** contains 85,474 referring expressions for 54,822 objects across 26,711 images, collected non-interactively. Notably, its descriptions are significantly more complex (8.43 words/expression) than RefCOCO (3.61) and RefCOCO+ (3.53). Additionally, RefCOCO and Ref-COCO+ exhibit higher intra-category object density (3.9 vs. 1.6 objects/image).

To evaluate generalization in multi-object and zero-object settings, we use the GRES benchmark and its gRefCOCO dataset. **gRefCOCO** contains 278,232 expressions, including 80,022 multi-target and 32,202 no-target expressions, referring to 60,287 distinct instances across 19,994 images; masks and boxes are provided for all targets, and a portion of single-target expressions are inherited from RefCOCO. Evaluation follows GRES with cIoU and gIoU (and separate no-target accuracy when relevant), using the same UNC partition as RefCOCO.

## A.2  COMPARISON

Although RefCOCO, RefCOCO+, and RefCOCOg share a common MSCOCO image base, they differ substantially in annotation style and linguistic complexity. RefCOCO features short, spatially grounded expressions (avg. 3.6 words) collected interactively. RefCOCO+ suppresses absolute location terms, encouraging models to rely on appearance-based cues. RefCOCOg contains longer, non-interactive descriptions (avg. 8.43 words), posing greater challenges in semantic parsing and low object density contexts. These differences introduce meaningful distribution shifts across both language and vision, making this setup a robust test for generalization.

# B  EVALUATION METRICS

Following the common practice (Yang et al., 2022; 2023; Zhang et al., 2022), we adopt the metrics of overall intersection-over-union (oIoU), mean intersection-over-union (mIoU), and P@X. Specifically, oIoU measures the total intersection area over the total union area across all test samples, while mIoU averages the per-sample IoU values. P@$X$ computes the percentage of samples with IoU exceeding threshold $X$, where $X \in \{0.5, 0.7, 0.9\}$. These metrics provide a comprehensive assessment of both global segmentation quality and alignment precision.

# C  TRAINING DETAILS

To ensure a fair comparison, we integrate AML into several representative RIS frameworks, including DETRIS (Huang et al., 2025) , CARIS (Liu et al., 2023c) and ReLA (Liu et al., 2023a). Unless otherwise specified, our main implementation is built upon CARIS (Liu et al., 2023c), a recent state-of-the-art method. We follow its original training protocol: the model is optimized using AdamW (Loshchilov & Hutter, 2017) with a weight decay of 0.01. The initial learning rate is set to 1e-5 for the image/text encoders and 1e-4 for the remaining components. A poly learning rate decay strategy is applied with a power parameter of 0.9, gradually annealing the learning rate to zero. All models are trained for 50 epochs with a batch size of 16. The input resolution is fixed to $448 \times 448$ for both training and testing. For the proposed AML framework, we adopt the following

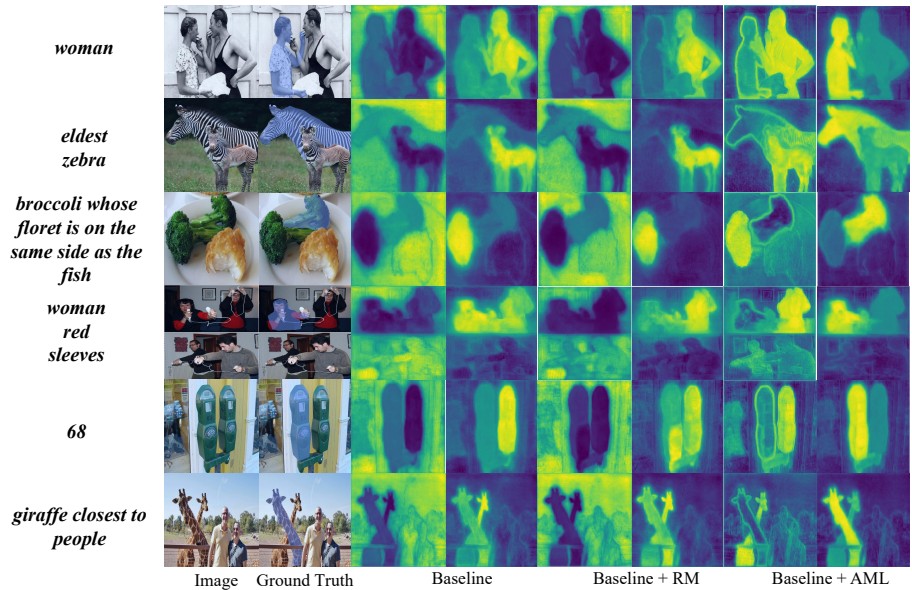

Figure 5: More comparison of prediction maps: Baseline , Random Mask (RM), and AML.

default hyperparameters: $\tau = 0.4, \rho = 0.25, H_P \times W_p = 32 \times 32, D_a = 2048$. All experiments are conducted on a single A800 GPU.

## D  MORE VISUALIZATION OF PREDICTION MAPS

The more comparison of prediction maps is shown as Figure 5, which consistently demonstrates our superior performance in both boundary precision and semantic discrimination.

## E  DETAILS FOR ANALYSIS OF CROSS-MODAL SIMILARITY

For the computation of $S$, our method directly uses the similarity scores from PMME, while for the baseline model, to ensure fair comparison, we replicate the identical projection layer and append it to features after encoding. Since the projection layer in PMME remains fixed after initialization, this approach introduces no unfairness in training adaptation. Our statistical analysis is conducted on 100 randomly batches sampled instances from RefCOCO+ val dataset, yielding a total of 313600 similarity scores. As shown in Figure 6, our method yields a higher mean similarity and a lower standard deviation, indicating more concentrated and semantically coherent feature activations. Additionally, the proportion of visual regions with strong similarity also improves substantially. These results demonstrate that our discrepancy-aware masking effectively suppresses misaligned or ambiguous regions, allowing the encoder to focus on high-confidence, text-relevant areas.

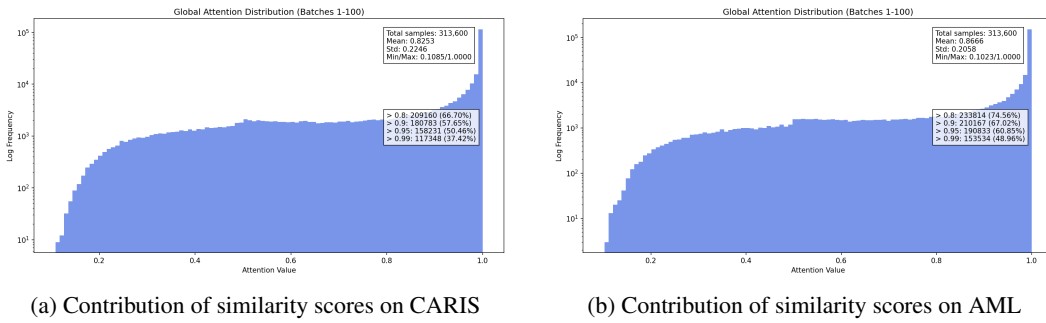

(a) Contribution of similarity scores on CARIS          (b) Contribution of similarity scores on AML

Figure 6: Contribution of similarity scores.

# F PROOF FOR PMME'S RANDOM PROJECTION ALIGNMENT

## F.1 THEOREM1: CROSS-MODAL INNER PRODUCT PRESERVATION VIA GAUSSIAN RANDOM MAPPING

Let $\varepsilon \in (0,1)$ be the distortion parameter and $\sigma \in (0,1)$ the target failure probability. Consider two sets of normalized feature vectors:

$$\{v_m \in \mathbb{R}^{D_i}\}_{m=1}^M, \quad \{u_n \in \mathbb{R}^{D_t}\}_{n=1}^N, \quad \text{where} \quad \|v_m\|_2 = \|u_n\|_2 = 1.$$

Define a block-diagonal random projection:

$$\widetilde{W} = \frac{1}{\sqrt{2}} \operatorname{diag}(W_i, W_t) \in \mathbb{R}^{(D_i + D_t) \times 2D_a},$$

where $W_i \in \mathbb{R}^{D_i \times D_a}$ and $W_t \in \mathbb{R}^{D_t \times D_a}$ are random matrices with entries $(W_i)_{kl}, (W_t)_{kl} \sim \mathcal{N}(0, \frac{1}{D_a})$ *i.i.d.* Define the projected embeddings:

$$v'_m = W_i^\top v_m \in \mathbb{R}^{D_a}, \quad u'_n = W_t^\top u_n \in \mathbb{R}^{D_a}.$$

If the projection dimension satisfies

$$D_a \geq \frac{8 \log(MN/\sigma)}{\varepsilon^2},$$

then with probability at least $1 - \sigma$, the following holds:

$$|\langle v'_m, u'_n \rangle - \langle v_m, u_n \rangle| \leq \varepsilon.$$

In other words, the inner products between all visual-language vector pairs are approximately preserved under the random block-diagonal projection.

## F.2 PROOF FOR THEOREM1

**Step 1: Construct the concatenated embedding space.** Define the joint embedding set:

$$\mathcal{Z} = \left\{ z_{mn} = \begin{bmatrix} v_m \\ u_n \end{bmatrix} \in \mathbb{R}^{D_v + D_t} \right\}_{m \in [M], \, n \in [N]}.$$

Let $\widetilde{W} = \frac{1}{\sqrt{2}} \operatorname{diag}(W_v, W_t) \in \mathbb{R}^{(D_v + D_t) \times D}$, where $W_v$ and $W_t$ are independent Gaussian random matrices with i.i.d. entries from $\mathcal{N}(0, 1/D)$.

**Step 2: Distance preservation over all pairwise combinations.** We wish to ensure that for all pairs $(m, n), (m', n')$:

$$(1 - \varepsilon)\|z_{mn} - z_{m'n'}\|^2 \leq \|\widetilde{W} z_{mn} - \widetilde{W} z_{m'n'}\|^2 \leq (1 + \varepsilon)\|z_{mn} - z_{m'n'}\|^2,$$

with high probability. According to **Theorem2** below, for a fixed pair of vectors, by standard results (e.g., chi-squared tail bound, Hanson–Wright inequality), we have:

$$\Pr\left[ \left| \|\widetilde{W} z - \widetilde{W} z'\|^2 - \|z - z'\|^2 \right| \geq \varepsilon \|z - z'\|^2 \right] \leq 2 \exp(-cD\varepsilon^2), \quad \text{for } z, z' \in \mathcal{Z}.$$

There are at most $\binom{MN}{2} \leq \frac{(MN)^2}{2}$ distinct pairs. Using a union bound and requiring total failure probability $\leq \delta$, we set:

$$2 \exp(-cD\varepsilon^2) \leq \frac{\delta}{(MN)^2},$$

which gives:

$$D \geq \frac{1}{c} \cdot \frac{\log(2(MN)^2/\delta)}{\varepsilon^2}.$$

Using $c = 1/8$ (standard for Gaussian), we simplify:

$$D \geq \frac{8 \log(MN/\delta)}{\varepsilon^2}.$$

Hence, with probability at least $1 - \delta$, the Euclidean distance between all projected pairs is preserved within $(1 \pm \varepsilon)$ distortion.

**Step 3: From distance preservation to inner product preservation.** Assume all $v_m$, $u_n$ are unit vectors. Then:

$$\|z_{mn}\|^2 = \|v_m\|^2 + \|u_n\|^2 = 2.$$

For any $(m, n), (m', n')$, define $z = z_{mn}$ and $z' = z_{m'n'}$. We use the identity:

$$\langle z, z' \rangle = 1 - \frac{1}{2}\|z - z'\|^2.$$

According to Theorem2 (see Appendix F.3), we know that:

$$(1 - \varepsilon)\|z - z'\|^2 \leq \|\widetilde{W}z - \widetilde{W}z'\|^2 \leq (1 + \varepsilon)\|z - z'\|^2.$$

Then:

$$\left| \langle \widetilde{W}z, \widetilde{W}z' \rangle - \langle z, z' \rangle \right| \leq \frac{1}{2}\left| \|\widetilde{W}z - \widetilde{W}z'\|^2 - \|z - z'\|^2 \right| \leq \frac{1}{2}\varepsilon \cdot \|z - z'\|^2.$$

Given $\|z - z'\| \leq 2$, we have

$$\left| \langle \widetilde{W}z, \widetilde{W}z' \rangle - \langle z, z' \rangle \right| \leq \frac{1}{2}\varepsilon \|z - z'\|^2 \leq 2\varepsilon.$$

Thus the geometric distortion between samples before and after projection is bounded by $2\varepsilon$, which is the foundation of our approximate modality-lifting guarantee.

### F.3   THEOREM2: BLOCK-DIAGONAL DISTANCE PRESERVATION

Let $\varepsilon \in (0, 1)$ be the distortion parameter, $\delta \in (0, 1)$ the target failure probability, and suppose we are given two modality-specific datasets consisting of $n$ paired samples:

$$\{(u_i, v_i)\}_{i=1}^n, \quad u_i \in \mathbb{R}^p, \quad v_i \in \mathbb{R}^q.$$

Define the concatenated vector $x_i = \begin{pmatrix} u_i \\ v_i \end{pmatrix} \in \mathbb{R}^{p+q}$, and consider a random block-diagonal projection:

$$\widetilde{W} = \frac{1}{\sqrt{2}} \operatorname{diag}(W_u, W_v),$$

where $W_u \in \mathbb{R}^{d \times p}$ and $W_v \in \mathbb{R}^{d \times q}$ have i.i.d. entries sampled from $\mathcal{N}(0, 1/d)$.

If the projection dimension $d$ satisfies

$$d \geq \frac{8 \log(n^2/\delta)}{\varepsilon^2},$$

then, with probability at least $1 - \delta$, the following inequality holds uniformly for all $\frac{n(n-1)}{2}$ distinct sample pairs $(i, j)$:

$$\boxed{(1 - \varepsilon)\|x_i - x_j\|_2^2 \leq \|\widetilde{W}x_i - \widetilde{W}x_j\|_2^2 \leq (1 + \varepsilon)\|x_i - x_j\|_2^2, \quad \forall 1 \leq i < j \leq n.}$$

**Interpretation.** This theorem ensures that all pairwise Euclidean distances between the joint embeddings $(u_i, v_i)$ are approximately preserved under the random block-diagonal projection $\widetilde{W}$. The guarantee holds uniformly with high probability as long as the projection dimension $d$ scales logarithmically with the number of pairs $n^2$ and inverse failure rate $1/\delta$.

### F.3.1 LEMMA 1: GAUSSIAN PROJECTION YIELDS SCALED $\chi^2$ DISTRIBUTION

We begin by analyzing the distribution of $\|Ww\|_2^2$ when $W \in \mathbb{R}^{d \times p}$ is a random Gaussian matrix with i.i.d. entries:

$$W_{k\ell} \sim \mathcal{N}\left(0, \frac{1}{d}\right), \quad \forall\, k = 1, \ldots, d;\ \ell = 1, \ldots, p,$$

and $w \in \mathbb{R}^p$ is any fixed vector.

Let $r_k \in \mathbb{R}^p$ denote the $k$-th row of $W$. Then the $k$-th component of the projected vector $Ww$ is:

$$\xi_k := r_k^\top w = \sum_{\ell=1}^p W_{k\ell} w_\ell.$$

By properties of Gaussian linear combinations, since the $W_{k\ell}$ are independent and $w$ is fixed, each $\xi_k$ is normally distributed:

$$\xi_k \sim \mathcal{N}\left(0, \sum_{\ell=1}^p \frac{1}{d} w_\ell^2\right) = \mathcal{N}\left(0, \frac{\|w\|_2^2}{d}\right).$$

Hence, the squared norm of the projected vector is:

$$\|Ww\|_2^2 = \sum_{k=1}^d \xi_k^2.$$

Now, observe that we can reparametrize each term as:

$$\frac{\xi_k}{\|w\|_2/\sqrt{d}} \sim \mathcal{N}(0,1), \quad \Rightarrow \quad \left(\frac{\xi_k}{\|w\|_2/\sqrt{d}}\right)^2 \sim \chi_1^2.$$

Summing over all $k = 1, \ldots, d$, we obtain:

$$\sum_{k=1}^d \left(\frac{\xi_k}{\|w\|_2/\sqrt{d}}\right)^2 \sim \chi_d^2.$$

Therefore, we conclude:

$$\boxed{\|Ww\|_2^2 = \frac{\|w\|_2^2}{d} \cdot \chi_d^2.}$$

This shows that the squared norm of the projected vector follows a scaled chi-squared distribution, which concentrates tightly around its expectation as $d$ grows.

### F.3.2 LEMMA 2: TAIL BOUND FOR $\chi_d^2$ VIA BERNSTEIN'S INEQUALITY

We now establish a concentration bound for the $\chi^2$ random variable appearing in Step F.3.1. The key result is the following:

**Result.** Let $\chi_d^2 = \sum_{i=1}^d X_i^2$, where $X_i \sim \mathcal{N}(0,1)$ i.i.d. Then, for any $\varepsilon \in (0,1)$,

$$\boxed{\Pr\left[\left|\chi_d^2 - d\right| \geq \varepsilon d\right] \leq 2\exp\left(-\frac{d\varepsilon^2}{8}\right).} \tag{A.2}$$

This is the tail bound used in the JL lemma analysis, and we now prove it by applying Bernstein's inequality.

**Step 1: Define the zero-mean variable.** Let

$$S := \chi_d^2 - d = \sum_{i=1}^{d} (X_i^2 - 1).$$

Note that $\mathbb{E}[S] = 0$.

We are interested in bounding $\Pr[|S| \geq \varepsilon d]$.

**Step 2: Verify Bernstein's inequality conditions.** Each $X_i^2 - 1$ is a centered sub-exponential random variable:

$$X_i^2 - 1 \in \mathrm{SubExp}(K), \quad \text{with } K = \|X_i^2 - 1\|_{\psi_1} < \infty,$$

as shown in Vershynin (2018, Lemma 2.7.7). The variance is:

$$\mathrm{Var}(X_i^2) = \mathbb{E}[X_i^4] - \left(\mathbb{E}[X_i^2]\right)^2 = 3 - 1 = 2.$$

Therefore, the sum $S$ satisfies:

$$\sigma^2 = \sum_{i=1}^{d} \mathrm{Var}(X_i^2 - 1) = 2d.$$

**Step 3: Apply Bernstein's inequality.** This inequality can be found in the Theorem 2.8.1 in Vershynin's Hihg-Dimentional Probability (Vershynin, 2009). Let $t = \varepsilon d$, then Bernstein gives:

$$\boxed{\Pr[S \geq t] \leq \exp\left(-\min\left(\frac{t^2}{4\sigma^2}, \frac{t}{2K}\right)\right) = \exp\left(-\min\left(\frac{(\varepsilon d)^2}{8d}, \frac{\varepsilon d}{2K}\right)\right).}$$

Since $\varepsilon \in (0,1)$ and $d$ is large, the first term dominates:

$$\Pr[S \geq \varepsilon d] \leq \exp\left(-\frac{d\varepsilon^2}{8}\right).$$

By symmetry, the lower tail satisfies the same bound:

$$\Pr[S \leq -\varepsilon d] \leq \exp\left(-\frac{d\varepsilon^2}{8}\right).$$

**Step 4: Combine upper and lower bounds.** We conclude:

$$\Pr\left[|\chi_d^2 - d| \geq \varepsilon d\right] = \Pr[|S| \geq \varepsilon d] \leq 2\exp\left(-\frac{d\varepsilon^2}{8}\right),$$

which completes the proof.

**Implication for Projection Norm.** Recall from Step F.3.1 that:

$$\|Ww\|_2^2 = \frac{\|w\|_2^2}{d} \cdot \chi_d^2,$$

therefore:

$$\boxed{\Pr\left[\left|\|Ww\|_2^2 - \|w\|_2^2\right| \geq \varepsilon\|w\|_2^2\right] = \Pr\left[|\chi_d^2 - d| \geq \varepsilon d\right] \leq 2\exp\left(-\frac{d\varepsilon^2}{8}\right).} \tag{A.3}$$

This tail inequality will be used in the next step to uniformly control all pairwise distortions.

### F.3.3 PROOF OF THEOREM 2

**Step 1: Formulation.** We now prove the distance-preservation guarantee for random block-diagonal projections over all pairs of concatenated visual-language embeddings. Recall we are given $n$ pairs $(u_i, v_i)$, where $u_i \in \mathbb{R}^p$ (visual) and $v_i \in \mathbb{R}^q$ (language). Define the concatenated embedding:

$$x_i = \begin{pmatrix} u_i \\ v_i \end{pmatrix} \in \mathbb{R}^{p+q}, \quad x_i - x_j = \begin{pmatrix} a_{ij} \\ b_{ij} \end{pmatrix}.$$

Let the random projection matrix be:

$$\widetilde{W} = \frac{1}{\sqrt{2}} \operatorname{diag}(W_u, W_v), \quad W_u \sim \mathcal{N}(0, 1/d)^{d \times p}, \; W_v \sim \mathcal{N}(0, 1/d)^{d \times q}.$$

**Step 2: Concentration for each pairwise difference.** From Lemma F.3.2, we know:

$$\Pr\left[\left| \|W_u a_{ij}\|^2 - \|a_{ij}\|^2 \right| \geq \varepsilon \|a_{ij}\|^2 \right] \leq \sigma/2,$$

and similarly for $\|W_v b_{ij}\|^2$. These bounds hold uniformly over all $a_{ij}, b_{ij}$ for fixed $(i, j)$, under the dimensionality condition:

$$d \geq \frac{8 \log(2n^2/\sigma)}{\varepsilon^2}.$$

**Step 3: Joint success probability.** Let event $A_{ij}$ be: $\|W_u a_{ij}\|^2 \in [(1-\varepsilon)\|a_{ij}\|^2, \; (1+\varepsilon)\|a_{ij}\|^2]$, and $B_{ij}$ be the corresponding event on $W_v$ side. Then:

$$\Pr[A_{ij}] \geq 1 - \frac{\sigma}{2}, \quad \Pr[B_{ij}] \geq 1 - \frac{\sigma}{2}.$$

Since $W_u$ and $W_v$ are independent, the joint event satisfies:

$$\Pr[A_{ij} \cap B_{ij}] \geq \left(1 - \frac{\sigma}{2}\right)^2 \geq 1 - \sigma.$$

This gives a sharper bound than simply using union bound.

**Step 4: Combine into distance bound.** Under event $A_{ij} \cap B_{ij}$, we have:

$$\|W_u a_{ij}\|^2 \in [(1-\varepsilon)\|a_{ij}\|^2, (1+\varepsilon)\|a_{ij}\|^2],$$
$$\|W_v b_{ij}\|^2 \in [(1-\varepsilon)\|b_{ij}\|^2, (1+\varepsilon)\|b_{ij}\|^2].$$

So the projected squared distance satisfies:

$$\|\widetilde{W}(x_i - x_j)\|^2 = \frac{1}{2}(\|W_u a_{ij}\|^2 + \|W_v b_{ij}\|^2) \in \left[(1 - \varepsilon)\|x_i - x_j\|^2, \; (1 + \varepsilon)\|x_i - x_j\|^2\right].$$

Hence, with probability $\geq 1 - \sigma$, the block-diagonal random projection preserves the pairwise distance between $x_i$ and $x_j$ within $(1 \pm \varepsilon)$.

**Step 5: Extend to all $\binom{n}{2}$ pairs.** Let $|\mathcal{P}| = \binom{n}{2} < n^2$. Apply a union bound over all such pairs:

$$\Pr[\exists\, (i,j), \text{ violation occurs}] \leq |\mathcal{P}| \cdot \sigma \leq n^2 \cdot \sigma.$$

Set this to be $\leq \delta$. Then:

$$\sigma = \frac{\delta}{n^2} \Rightarrow d \geq \frac{8 \log(2n^2/\delta)}{\varepsilon^2}.$$

**Step 6: Final conclusion.** If $d \geq \frac{8 \log(2n^2/\delta)}{\varepsilon^2}$, then with probability at least $1 - \delta$, we have:

$$\boxed{(1 - \varepsilon)\|x_i - x_j\|^2 \leq \|\widetilde{W} x_i - \widetilde{W} x_j\|^2 \leq (1 + \varepsilon)\|x_i - x_j\|^2, \quad \forall i, j.}$$

# G ADDITIONAL EXPERIMENTAL RESULTS

## G.1 ABLATION: PROJECTION DIMENSION AND JL ERROR BOUND

To quantify the impact of the random projection dimension on cross-modal alignment and segmentation performance, we conducted a comparison on RefCOCO+ with projection dimensions $D_a \in \{1024, 2048, 4096\}$. All other settings remained consistent.

Table 8: Performance of different projection dimensions $D_a$ and corresponding JL error bounds.

| Dim ($D_a$) | oIoU | mIoU | Pr@50 | Pr@90 | JL Error Bound ($\epsilon$) |
|---|---|---|---|---|---|
| 1024 | 66.82 | 70.83 | 80.15 | 36.27 | 0.2955 |
| 2048 | 67.37 | 71.33 | 80.82 | 36.35 | 0.2088 |
| 4096 | 67.08 | 71.21 | 80.85 | 36.22 | 0.1478 |

As shown in Table 8, when the dimension $D_a$ increases from 1024 to 2048, we observe a noticeable improvement in all metrics: oIoU increases from 66.82 to 67.37, mIoU improves from 70.83 to 71.33, Pr@50 increases from 80.15 to 80.82, and Pr@90 rises from 36.27 to 36.35. However, further increasing the dimension to 4096 results in marginal gains, with oIoU reaching 67.08, mIoU at 71.21, Pr@50 at 80.85, and Pr@90 at 36.22.

Meanwhile, the theoretical error bound $\epsilon$ decreases monotonically with increasing $D_a$ (0.2955 → 0.2088 → 0.1478), which is consistent with the empirical trend. This confirms that higher dimensions better preserve cross-modal geometric structures. Based on the trade-off between accuracy and scalability, we choose $D_a = 2048$ as the default setting. This configuration achieves a good balance between controlled theoretical distortion bound ($\epsilon = 0.2088$) and diverse, semantically consistent region selection, while nearing saturation on the main segmentation metrics. Thus, this validates the rationale and cost-effectiveness of using this projection dimension in PMME.

## G.2 TRAINING OVERHEAD OF AML

We conducted an ablation study to quantify the additional training time and memory overhead introduced by the AML approach, specifically due to the two forward passes per iteration. We compared the models with and without AML under identical batch sizes. All other network components and training settings remained the same to ensure a fair comparison.

As shown in Table 9, we observe that the introduction of AML results in only a minimal increase in memory consumption (+4.9%) and training time per epoch (+17.2%). On DETRIS, the overhead is also limited to 5.3% in memory and 19.6% in time. These results suggest that the additional overhead introduced by AML is lightweight, especially when compared to the significantly larger overheads that would arise from incorporating additional data during training. These findings validate that the overhead introduced by AML is relatively small, supporting its efficiency while maintaining strong model performance.

Table 9: Comparison of training overhead.

| Method | Batch Size | Memory(MB) | Time/Epoch(min) |
|---|---|---|---|
| CARIS | 16 | 35,168 | 30.16 |
| CARIS+AML | 16 | 36,902(**+4.9%**) | 35.35(**+17.2%**) |
| DETRIS | 16 | 12,206 | 27.03 |
| DETRIS+AML | 16 | 12,860 (**+5.3%**) | 32.34 (**+19.6%**) |

We further analyze the training overhead and fairness of CARIS+AML compared with CARIS. While CARIS+AML introduces a 17.2% increase in training time due to an additional forward pass in the first stage, the total number of optimization steps remains unchanged, since parameter updates occur only in the second stage and each epoch sees the same number of batches as CARIS. As shown in Table 10, under the same wall-clock time (8/26/42 epochs for CARIS+AML vs. 10/30/50 for CARIS), AML consistently outperforms the baseline (e.g., 66.7/70.8 vs. 65.5/69.3 at the 50-epoch

time budget), and already reaches the 50-epoch CARIS performance using roughly the 30-epoch time budget (65.3/69.5 vs. 65.5/69.3), confirming that the improvements come from the alignment-aware masking strategy rather than from additional optimization steps.

Table 10: Analysis of training overhead and fairness on CARIS. Memory and time are measured with batch size 16. Performance is reported as mIoU / oIoU on RefCOCO+.

| Method | Memory (MB) | Time/Epoch (min) | 10 epochs | 30 epochs | 50 epochs |
|---|---|---|---|---|---|
| CARIS | 35,168 | 30.16 | 60.2 / 62.3 | 64.7 / 68.3 | 65.5 / 69.3 |
| +AML / same time | 36,902 (**+4.9%**) | 35.35 (**+17.2%**) | 61.8 / 64.2 | 65.3 / 69.5 | 66.7 / 70.8 |
| +AML / same steps | | | 63.8 / 65.4 | 66.1 / 70.1 | 67.4 / 71.3 |

### G.3 GENERALIZATION CAPABILITY FOR MULTI-TARGET AND COMPOSITIONAL EXPRESSIONS

The AML strategy is not limited to single-target referring expression localization; it is equally applicable to multi-target and compositional relational expressions. Specifically, the PatchMax Matching Estimation (PMME) in AML computes fine-grained similarities between all patches in the image and all tokens in the referring expression. Any patch that exhibits strong alignment with any token—whether it is a referring term (e.g., "dog") or contextual cues (e.g., "on the left", "bench")—is retained. This token-level matching mechanism preserves both the target regions and contextual regions necessary for disambiguation, which is particularly crucial for relational expressions.

For instance, consider the expression "the giraffe closest to that man". Correct localization depends on spatial relationships rather than appearance alone. AML is capable of preserving semantically critical regions (such as "man"), even without explicit multi-mask supervision.

To further evaluate AML's generalization capability in multi-target and zero-target scenarios, we integrate AML with ReLA and conduct experiments on the GRES dataset. The results are as follows:

Table 11: Performance comparison on GRES dataset.

| Method | cIoU@5k | gIoU@5k | cIoU@150k | gIoU@150k |
|---|---|---|---|---|
| ReLA | 28.49 | 17.33 | 62.50 | 63.25 |
| ReLA + AML | **30.86** | **21.08** | **62.61** | **63.52** |

Compared to ReLA, AML brings significant improvements in the early training stage: at 5k iterations, cIoU and gIoU improve by +2.37% and +3.75%, respectively. However, at 150k iterations, the improvement is marginal (+0.19%). This indicates that AML helps filter out misaligned regions in the early stage, but in multi-target or zero-target scenarios, its potential is not fully realized due to insufficient consistency in boundary definition. Future work will explore soft or adaptive masking strategies to better handle these complex scenarios.

### G.4 ALIGNMENT AND MASKING STABILITY ANALYSIS

Table 12: Evolution of cross-modal alignment metrics across training epochs 1–10. Arrows indicate desired optimization direction: ↓ for minimization, ↑ for maximization.

| Metric / Epoch | 1 | 2 | 3 | 4 | 5 | 6 | 7 | 8 | 9 | 10 |
|---|---|---|---|---|---|---|---|---|---|---|
| $\text{sim} < 0.20$ ↓ | 0.0041 | 0.0010 | 0.0008 | 0.0007 | 0.0008 | 0.0008 | 0.0007 | 0.0008 | 0.0007 | 0.0008 |
| $\text{sim} \geq 0.50$ ↑ | 0.8021 | 0.8464 | 0.8620 | 0.8656 | 0.8650 | 0.8700 | 0.8720 | 0.8769 | 0.8836 | 0.8863 |
| $\text{sim} \geq 0.90$ ↑ | 0.4192 | 0.4543 | 0.4667 | 0.4779 | 0.5022 | 0.5079 | 0.5076 | 0.5178 | 0.5196 | 0.5252 |
| mean(sim) ↑ | 0.7542 | 0.7821 | 0.7917 | 0.7963 | 0.8031 | 0.8061 | 0.8142 | 0.8117 | 0.8182 | 0.8211 |
| std(sim) ↓ | 0.2393 | 0.2230 | 0.2171 | 0.2161 | 0.2097 | 0.2075 | 0.2051 | 0.2044 | 0.2053 | 0.2041 |

We investigate the stability dynamics of our AML training strategy, which employs a curriculum-based alignment approach rather than enforcing uniform alignment during early training phases. The Alignment-aware Filtering Mask (AFM) progressively focuses on regions with high optimization

Table 13: Mask-on-ground-truth coverage rates across different threshold configurations during epochs 0–10. Lower values indicate better masking precision.

| $\tau$ / Epoch | 0 | 1 | 2 | 3 | 4 | 5 | 6 | 7 | 8 | 9 | 10 |
|---|---|---|---|---|---|---|---|---|---|---|---|
| 0.20 | 0.0021 | 0.0007 | 0.0004 | 0.0005 | 0.0005 | 0.0005 | 0.0005 | 0.0005 | 0.0004 | 0.0005 | 0.0004 |
| 0.30 | 0.0097 | 0.0057 | 0.0058 | 0.0053 | 0.0052 | 0.0051 | 0.0049 | 0.0044 | 0.0046 | 0.0043 | 0.0043 |
| 0.40 | 0.0259 | 0.0178 | 0.0157 | 0.0139 | 0.0135 | 0.0129 | 0.0130 | 0.0125 | 0.0121 | 0.0122 | 0.0120 |
| 0.60 | 0.0726 | 0.0628 | 0.0571 | 0.0559 | 0.0558 | 0.0543 | 0.0537 | 0.0526 | 0.0523 | 0.0519 | 0.0520 |

Table 14: Training loss comparison between CARIS baseline and AML across 50 epochs.

| Early Training Phase (Epochs 1–17) | | | | | | | | | |
|---|---|---|---|---|---|---|---|---|---|
| **Epoch** | **1** | **3** | **5** | **7** | **9** | **11** | **13** | **15** | **17** |
| CARIS | 0.3114 | 0.2060 | 0.1636 | 0.1413 | 0.1237 | 0.1109 | 0.1002 | 0.0919 | 0.0846 |
| AML | **0.2903** | **0.1803** | **0.1434** | **0.1209** | **0.1060** | **0.0928** | **0.0816** | **0.0742** | **0.0669** |
| Improvement | 6.8% | 12.5% | 12.3% | 14.4% | 14.3% | 16.3% | 18.6% | 19.3% | 20.9% |
| Mid Training Phase (Epochs 18–34) | | | | | | | | | |
| **Epoch** | **18** | **20** | **22** | **24** | **26** | **28** | **30** | **32** | **34** |
| CARIS | 0.0819 | 0.0759 | 0.0702 | 0.0643 | 0.0613 | 0.0565 | 0.0532 | 0.0501 | 0.0468 |
| AML | **0.0650** | **0.0580** | **0.0531** | **0.0489** | **0.0462** | **0.0424** | **0.0405** | **0.0385** | **0.0362** |
| Improvement | 20.6% | 23.6% | 24.4% | 23.9% | 24.6% | 25.0% | 23.9% | 23.2% | 22.6% |
| Late Training Phase (Epochs 35–50) | | | | | | | | | |
| **Epoch** | **35** | **37** | **39** | **41** | **43** | **45** | **47** | **49** | **50** |
| CARIS | 0.0458 | 0.0441 | 0.0415 | 0.0404 | 0.0382 | 0.0368 | 0.0352 | 0.0348 | 0.0342 |
| AML | **0.0348** | **0.0336** | **0.0321** | **0.0304** | **0.0297** | **0.0289** | **0.0281** | **0.0273** | **0.0271** |
| Improvement | 24.0% | 23.8% | 22.7% | 24.8% | 22.3% | 21.5% | 20.2% | 21.6% | 20.8% |

feasibility and internal consistency through threshold and masking ratio control, establishing stable alignment baselines before extending to ambiguous regions. This curriculum strategy mitigates instability issues inherent in unpaired vision-language backbones during initial training stages.

**Experimental Setup.** We evaluate alignment and masking stability using patch-text similarity measured by PMME and mask-on-ground-truth coverage rates. Experiments are conducted on a fixed evaluation set of 1,552 image-text pairs with $14 \times 14$ patch granularity. We track alignment metrics across epochs 1–10, including weak alignment ratio (sim $< 0.20$), moderate alignment ratio (sim $\geq 0.50$), high-confidence alignment ratio (sim $\geq 0.90$), mean similarity, and standard deviation. Mask accuracy is assessed by computing coverage rates of ground-truth alignment patches filtered by threshold $\tau \in \{0.20, 0.30, 0.40, 0.60\}$ during epochs 0–10. Convergence analysis compares 50-epoch training loss trajectories between CARIS baseline and AML-enhanced models under identical configurations.

**Alignment Dynamics.** Table 12 presents the evolution of cross-modal alignment metrics, revealing systematic improvement patterns. Weak alignment regions (sim $< 0.20$) exhibit dramatic reduction from 0.41% to 0.08% by epoch 10, with the most significant drop occurring between epochs 1–2 (0.41% $\rightarrow$ 0.10%). This rapid pruning of low-quality correspondences stabilizes thereafter, maintaining consistently low levels ($< 0.1\%$). Moderate alignment ratios demonstrate steady growth from 80.21% to 88.63%, while high-confidence alignments increase substantially from 41.92% to 52.52%, representing a 25.3% relative improvement. The similarity distribution undergoes concurrent upward shift (mean: $0.7542 \rightarrow 0.8211$, $+8.9\%$) and concentration (std: $0.2393 \rightarrow 0.2041$, $-14.7\%$), manifesting the characteristic "upward shift + convergence" pattern that validates AML's selective filtering mechanism.

**Masking Precision.** Table 13 analyzes mask-on-GT coverage across different threshold configurations, revealing threshold-dependent filtering precision. Conservative thresholds demonstrate excep-

tional accuracy: $\tau = 0.20$ maintains coverage below 0.1% throughout training ($0.21\% \rightarrow 0.04\%$), while $\tau = 0.30$ achieves 55.7% reduction ($0.97\% \rightarrow 0.43\%$). The $\tau = 0.40$ setting, despite higher initial coverage (2.59%), converges to acceptable levels (1.20%) representing 53.7% improvement. These results indicate effective preservation of critical positive signals. However, aggressive threshold $\tau = 0.60$ exhibits concerning behavior with coverage remaining elevated above 5%, suggesting potential over-masking that could impair learning signals despite 28.4% reduction from initial 7.26%.

**Convergence Characteristics.** Table 14 demonstrates superior optimization dynamics of AML across the complete 50-epoch training trajectory. Initial acceleration is pronounced, with first-epoch losses of 0.2903 (AML) versus 0.3114 (CARIS), representing 6.8% improvement. The advantage persists throughout training phases: early stage (epochs 1–17) shows consistent 7–15% loss reduction, mid-stage (epochs 18–34) maintains 15–23% advantage, and late-stage convergence (epochs 35–50) achieves 21–24% improvement. Terminal performance demonstrates 20.8% reduction in final loss (0.0271 vs. 0.0342), indicating significant steady-state error minimization alongside accelerated convergence. Notably, the loss gap widens progressively, suggesting AML's enhanced capacity for fine-grained optimization in later training phases.

The comprehensive quantitative analysis across alignment distribution evolution, masking precision metrics, and convergence dynamics provides robust evidence for AML's stabilization effects during early training phases. The systematic improvements in alignment quality (25.3% increase in high-confidence regions), masking accuracy (sub-3% coverage for conservative thresholds), and convergence efficiency (20.8% final loss reduction) substantiate the effectiveness of curriculum-based alignment strategies in addressing fundamental challenges of alignment and masking instability under unpaired backbone architectures.

### G.5 GENERALIZATION AND ROBUSTNESS ANALYSIS

To comprehensively evaluate the model's real-world applicability and robustness, we designed a series of rigorous tests. This is particularly crucial for the Referring Image Segmentation (RIS) task, as the model must parse diverse and complex natural language queries. For instance, the model might be required to locate an object based on its spatial position (e.g., *"the car in the bottom right corner"*, *"the left skier in the foreground"*), its attributes and actions (e.g., *"a fat woman playing with a frisbee"*), or its relationship with surrounding objects (e.g., *"the banana separated from the bunch"*).

A truly robust model must not only accurately interpret these expressions under ideal image conditions but also maintain stable performance when the visual input is severely degraded. Therefore, we introduce seven simulated visual transformations to systematically evaluate the model's ability to tackle this challenge. Visualizations of these transformations are shown in Figure 7. These transformations and their objectives are as follows:

- **Haze**: By overlaying a semi-transparent white mask with $50\%$ opacity, we simulate foggy or hazy weather conditions. This transformation reduces overall image contrast and clarity, testing the model's ability to identify objects in low-visibility scenarios.

- **Highlight**: The overall brightness is increased by a fixed factor of $1.5\times$ to simulate scenes under intense light or overexposure. This can lead to a loss of texture and detail in bright areas of the image.

- **Lowlight**: The overall brightness is decreased by a fixed factor of $0.45\times$ to simulate scenes with insufficient light, at night, or underexposure. This makes details in dark areas of the image difficult to discern.

- **Contrast**: The intensity difference between light and dark areas is enhanced by a fixed factor of $1.8\times$. By increasing the image contrast, we challenge the model's ability to discern object edges and textures.

- **Occlusion Box**: An opaque gray box with side length equal to $20\%$ of the image's shorter dimension is placed at a random location on the image to simulate real-world scenarios where the target object is partially occluded. This forces the model to infer and segment the complete outline of an object from its visible parts.

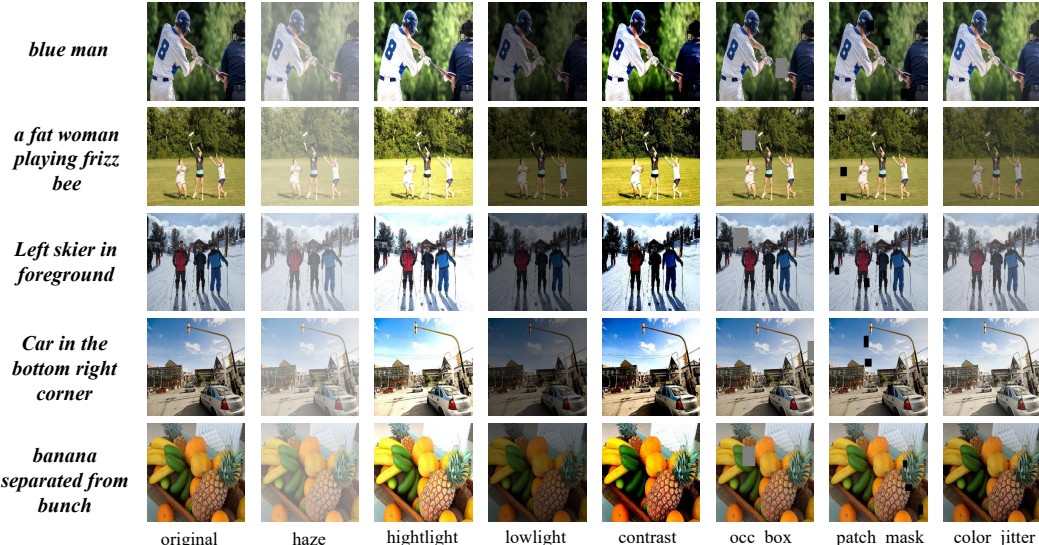

Figure 7: Visualization of the seven visual transformations used to simulate diverse visual conditions. From left to right: Original, Haze, Highlight, Lowlight, Contrast, Occlusion Box, Patch Masking, and Color Jittering.

- **Patch Masking**: The image is randomly overlaid with three black rectangular blocks of varying sizes and positions to simulate scattered multi-point information loss. This represents a more severe, structured loss of information than a single occlusion box, testing the model's ability to leverage global context for local content inference.

- **Color Jittering**: The image's brightness, contrast, and saturation are randomly altered within the range $[0.6, 1.4]$. This simulates color distortions caused by different camera sensors, varying lighting conditions, or image post-processing, testing the model's generalization to color variations.

Through this series of comprehensive evaluations, we can more reliably measure the AML strategy's combined strength in handling linguistic diversity and visual uncertainty. The performance results are presented in Table 15.

Table 15: Cross-dataset robustness under various visual transformations. We train on RefCOCO+ and evaluate on the validation splits of RefCOCO and RefCOCOg. Results are reported as mIoU/oIoU/pr@50. Our method (AML) consistently outperforms the baseline.

| Transformation | RefCOCO | | RefCOCOg | |
| --- | --- | --- | --- | --- |
| | Baseline | Ours (AML) | Baseline | Ours (AML) |
| Haze | 59.33 / 57.85 / 70.18 | 62.54 / 60.63 / 73.32 | 51.23 / 48.69 / 58.66 | 53.38 / 50.60 / 60.99 |
| Highlight | 57.81 / 59.65 / 69.76 | 62.28 / 59.85 / 72.76 | 51.74 / 48.98 / 59.74 | 53.66 / 50.83 / 61.50 |
| Lowlight | 58.32 / 56.81 / 69.07 | 61.63 / 59.81 / 72.46 | 50.46 / 48.02 / 58.03 | 52.07 / 49.43 / 60.13 |
| Contrast | 60.05 / 58.15 / 70.29 | 63.05 / 60.58 / 73.62 | 52.03 / 49.09 / 59.82 | 54.68 / 51.83 / 62.85 |
| Occlusion Box | 57.20 / 55.78 / 67.23 | 61.94 / 60.26 / 72.62 | 49.40 / 47.20 / 56.47 | 53.11 / 50.89 / 60.99 |
| Patch Masking | 60.22 / 58.51 / 70.74 | 63.18 / 61.17 / 74.04 | 52.71 / 50.02 / 60.92 | 54.71 / 51.73 / 62.62 |
| Color Jitter | 59.93 / 58.27 / 70.58 | 62.71 / 60.58 / 73.33 | 51.94 / 49.25 / 59.84 | 54.25 / 51.23 / 62.17 |
| **Average** | **58.98 / 57.86 / 69.69** | **62.48 / 60.41 / 73.16** | **51.36 / 48.75 / 59.07** | **53.69 / 50.93 / 61.61** |
| **Improvement** | — | **+3.50 / +2.55 / +3.47** | — | **+2.34 / +2.18 / +2.54** |

As shown in Table 15, AML consistently outperforms the baseline across all visual transformations. On RefCOCO, the average improvements are +3.45% for mIoU, +2.53% for oIoU, and +3.49% for pr@50. On RefCOCOg, the average improvements are +2.32% for mIoU, +2.11% for oIoU, and +2.61% for pr@50. The consistent improvements under different expression styles and severe visual degradation conditions demonstrate that AML possesses strong transferability and robustness beyond the training distribution.

Furthurmore, we conducted visual experiments on the Flickr30k dataset by loading the model trained on RefCOCO+ and performing inference on single-object images from Flickr30k. The results, as shown in the Figure 8, demonstrates that our method maintains precise decision boundaries even on data outside of RefCOCO.

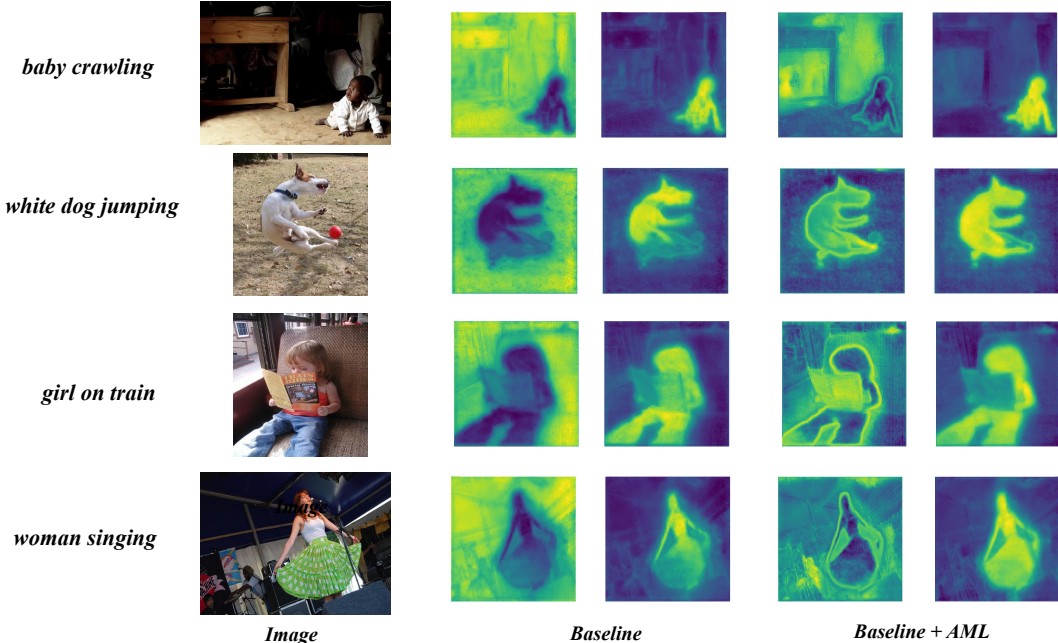

Figure 8: Visual segmentation results on the Flickr30k dataset, where models trained on RefCOCO+.

### G.6    DETAILED ABLATION STUDY ON HYPERPARAMETERS $\tau$ AND $\rho$

In this section, we present a comprehensive ablation study on two key hyperparameters of our AML strategy: the masking threshold $\tau$ and the dropout ratio $\rho$. This study aims to validate the robustness of our method to these critical hyperparameters and to provide an empirical basis for the parameter settings adopted in our main experiments.

### G.6.1    ANALYSIS OF DROPOUT RATIO $\rho$

The dropout ratio $\rho$ defines the proportion of pixels, among those marked as "misaligned" by $\tau$, that are randomly set to zero. To investigate its impact, we tested various $\rho$ values under two well-performing threshold settings ($\tau = 0.3$ and $\tau = 0.4$). These experiments were conducted on the RefCOCO+ testB set, with results summarized in Table 16.

The analysis of Table 16 further solidifies the effectiveness and robustness of our method. First, all tested $(\tau, \rho)$ combinations significantly outperform the baseline, with average performance gains ranging from +0.62 to +2.52. This strongly demonstrates the general effectiveness of our AML strategy, as its benefits are not contingent on a single, hard-to-find parameter setting. Second, examining both threshold configurations reveals that $\rho = 1/4$ consistently yields optimal performance across different $\tau$ values, achieving the highest average gains of +2.52 and +2.09 for $\tau = 0.3$ and $\tau = 0.4$, respectively. This dropout ratio achieves an effective balance: higher ratios ($\rho = 1/2$) may over-suppress learning signals from moderately aligned regions, while lower ratios ($\rho = 1/6$) may be insufficient in filtering noise from misaligned areas. The relatively stable performance of $\rho = 0.25$ across multiple threshold configurations suggests that retaining approximately 75% of misaligned pixels can provide appropriate regularization while preserving contextual information necessary for cross-modal reasoning. Moreover, this consistent superior performance across multiple threshold configurations provides direct and compelling empirical support for our choice to fix $\rho = 0.25$ in the main paper and in the analysis of Subsection G.6.2.

Table 16: Performance on RefCOCO+ testB with different dropout ratios $\rho$. We report individual metrics, their average (Avg. Metric), and the average gain over the baseline.

| Setting | mIoU | oIoU | Pr@50 | Pr@70 | Pr@90 | Avg. Metric | Gain over Baseline |
|---|---|---|---|---|---|---|---|
| Baseline | 62.69 | 57.97 | 70.24 | 62.41 | 31.27 | 56.92 | — |
| $\tau = 0.3, \rho = 1/2$ | 63.79 | 59.19 | 71.69 | 63.69 | 33.42 | 58.36 | +1.44 |
| $\tau = 0.3, \rho = 1/4$ | 64.82 | 59.74 | 72.65 | 65.23 | 34.77 | 59.44 | +2.52 |
| $\tau = 0.3, \rho = 1/6$ | 64.61 | 59.41 | 72.18 | 64.41 | 34.73 | 59.07 | +2.15 |
| $\tau = 0.4, \rho = 1/2$ | 64.15 | 58.95 | 72.08 | 63.98 | 34.77 | 58.59 | +1.67 |
| $\tau = 0.4, \rho = 1/4$ | 64.61 | 59.51 | 72.04 | 64.63 | 34.26 | 59.01 | +2.09 |
| $\tau = 0.4, \rho = 1/6$ | 63.44 | 58.75 | 70.81 | 62.88 | 31.83 | 57.54 | +0.62 |

### G.6.2 ANALYSIS OF MASKING THRESHOLD $\tau$

The masking threshold $\tau$ determines which visual regions are filtered based on their alignment with the text. A suitable value for $\tau$ must strike a balance between preserving useful information and filtering out noisy regions. To independently evaluate the impact of $\tau$, we first fix the dropout ratio $\rho = 0.25$, a value demonstrated to perform optimally(see Subsection G.6.1). We then test the performance across multiple benchmark datasets as $\tau$ varies within the range [0.2, 0.4]. The results are presented in Table 17.

Table 17: Performance with different threshold $\tau$ values across multiple datasets. Best performance on each dataset is **bolded**.

| Threshold $\tau$ | RefCOCO | | RefCOCO+ | | RefCOCOg |
|---|---|---|---|---|---|
| | val | testA | val | testA | val |
| 0.2 | 75.00 | 78.14 | 66.99 | 58.05 | **65.78** |
| 0.3 | **75.81** | 78.09 | 66.82 | 59.19 | 65.13 |
| 0.4 | 75.45 | **78.33** | **67.37** | **59.51** | 65.67 |

Several key conclusions can be drawn from the results in Table 17. First, the necessity of a non-zero threshold is evident. As mentioned in the main text, with $\tau = 0$ (i.e., no filtering), the performance on RefCOCO and RefCOCO+ is 74.65 and 65.54, respectively. All configurations with $\tau > 0$ in Table 17 outperform this baseline, demonstrating that filtering poorly aligned pixel regions is crucial for guiding the model's optimization.

Second, our method exhibits remarkable performance stability and robustness for $\tau \in [0.2, 0.4]$. The performance does not fluctuate drastically with changes in $\tau$, indicating that our approach provides benefits without requiring tedious fine-tuning. We also explored a dynamic thresholding strategy, where $\tau$ was adjusted per sample based on prediction confidence and referring expression length. However, this complex approach did not yield superior results compared to a fixed $\tau = 0.4$.

Considering the robust performance across all datasets, particularly on the more challenging Ref-COCO+ and RefCOCOg, we select $\tau = 0.4$ as our default value for the main experiments, as it offers an excellent balance between aggressive filtering and stable performance.

### G.7 ABLATION: IMAGE-LEVEL VS. FEATURE-LEVEL MASKING

To validate the effectiveness of our image-level alignment-aware masked (AML) approach, we conduct a comprehensive ablation study comparing different masking strategies. Masking mechanisms can be implemented at two distinct levels: feature-level and image-level. Existing methods such as Mask2Former (Cheng et al., 2022) typically adopt feature-level masking, applying spatial suppression during feature interaction processes. In contrast, our method performs masking operations at the image level, i.e., removing low-alignment regions before feature extraction. To comprehensively evaluate the effectiveness of these two strategies, we design corresponding comparative experiments with the following setup:

Table 18: Comparison of different masking strategies on RefCOCO+ dataset using oIoU metric.

| Method | val | testA | testB | Avg. |
|---|---|---|---|---|
| Baseline | 65.54 | 71.86 | 57.97 | 65.12 |
| + Feature Mask | 66.34 (+0.80) | 71.21 (-0.65) | 58.32 (+0.35) | 65.29 (+0.17) |
| + AML | **67.37 (+1.83)** | **73.19 (+1.33)** | **59.51 (+1.54)** | **66.69 (+1.57)** |

**Feature-level Masking.** Following standard practices in segmentation literature (Cheng et al., 2022) , we implement a feature-level variant of AML where alignment masks are applied during feature interaction processes. Specifically, the similarity map $S$ is computed on visual encoder features $F_v \in \mathbb{R}^{H \times W \times D}$, and low-alignment regions are suppressed through attention gating in downstream modules. The feature-level gating mechanism is formulated as:

$$\text{Attention}(Q, K, V) = \text{SoftMax}\left(\frac{QK^T + M_{\text{mask}}}{\sqrt{d_k}}\right) V \tag{16}$$

where the spatial mask $M_{\text{mask}} \in \mathbb{R}^{H \times W}$ is defined as:

$$M_{\text{mask}}[i, j] = \begin{cases} 0 & \text{if } S[i, j] > \tau \\ -\infty & \text{otherwise} \end{cases} \tag{17}$$

Here, $M_{\text{mask}}$ is a spatial mask used to suppress invalid positions before the softmax operation.

**Image-level Masking (Ours).** In contrast, our image-level AML removes misaligned regions before feature extraction, ensuring that only vision-language consistent content enters the visual encoder. Results are shown in table 18

**Results demonstrate that image-level masking significantly outperforms feature-level methods.** First, our method achieves consistent improvements across all evaluation splits, with an average gain of 1.57 oIoU points, demonstrating the robustness of our approach. In contrast, while feature-level masking provides marginal improvements (average +0.17), it exhibits unstable performance across different splits, even showing performance degradation on testA. This difference stems from the fundamental design distinctions between the two approaches: image-level masking prevents partial noise propagation through early intervention, while feature-level masking allows low-alignment regions to participate in early self-attention computations, leading to noise propagation through encoder layers before suppression. These results validate the effectiveness of early intervention strategies and provide important guidance for masking strategy selection in vision-language tasks.

## H    USE OF LARGE LANGUAGE MODELS

A large language model (LLM) was used *only* for stylistic polishing of prose (grammar, clarity) and for assisting table formatting (e.g., column alignment, caption phrasing). All content was authored, verified, and is fully owned by the authors, who take full responsibility for the submission.

