# OpenReview forum: "AMLRIS: Alignment-aware Masked Learning for Referring Image Segmentation"
_ICLR.cc/2026/Conference — ICLR 2026 Poster_

### Official Review · Reviewer_X2d4 · 2025-10-19

**Soundness:** 2
**Presentation:** 2
**Contribution:** 2
**Rating:** 4
**Confidence:** 5

**Summary:**

The paper proposes Alignment-Aware Masked Learning (AML) for referring image segmentation (RIS). AML computes a patch–token alignment map via PMME (random projections per modality) and then applies an Alignment-Aware Filtering Mask (AFM) that suppresses low-alignment image regions during training only; inference is unchanged. Experiments show consistent gains on RefCOCO/+/g when AML is added to CARIS, smaller gains on DETRIS, early-stage experiments on ReLA, and robustness to synthetic perturbations.

**Strengths:**

1.  **Plug-and-Play Method with Zero Inference Overhead** - Masking is applied exclusively during training. Consequently, the method requires no architectural changes and incurs no additional compute or memory costs during inference.

2.  **Strong Performance** - AML consistently boosts CARIS performance across all eight RefCOCO/RefCOCO+/RefCOCOg splits. It also transfers effectively to other baselines, such as DETRIS, and shows promising results with ReLA.

3.  **Robustness and Interpretability Analysis** - The paper shows improved performance under seven different visual perturbations (e.g., haze, low-light, occlusion), supported by clear qualitative maps that explain where AML helps.

4.  **Thorough Ablation Studies** - The paper contains comprehensive ablations examining the threshold, projection dimension, projection strategy (random vs. learnable), and masking level (image vs. feature), which validate the design choices.

**Weaknesses:**

1. **Gaps in Mathematical Derivations.** The appendix asserts
   $$
   \langle W_i v, W_t u\rangle = \tfrac{1}{2}\langle \tilde W z, \tilde W z\rangle,\quad
   \tilde W = \tfrac{1}{\sqrt{2}}\mathrm{diag}(W_i, W_t),\quad
   z = \begin{bmatrix} v \ u \end{bmatrix}.
   $$
   But
   $$
   \langle \tilde W z, \tilde W z\rangle
   = \left\langle \tfrac{1}{\sqrt{2}}\begin{bmatrix} W_i v \ W_t u \end{bmatrix},
   \tfrac{1}{\sqrt{2}}\begin{bmatrix} W_i v \ W_t u \end{bmatrix}\right\rangle
   = \tfrac{1}{2}\big(|W_i v|^2 + |W_t u|^2\big),
   $$
   which is the *sum of squared norms* (no cross term), so it cannot equal $\langle W_i v, W_t u\rangle$ except in degenerate cases. Moreover, with *independent* Gaussian projections $W_i, W_t$ (entries $\sim \mathcal N(0, 1/D_a)$) and unit $v, u$,
   $$
   \mathbb{E}[\langle W_i v, W_t u\rangle] = 0,\qquad
   \mathrm{Var}[\langle W_i v, W_t u\rangle] = \tfrac{1}{D_a},
   $$
   so the projected cross dot *concentrates at 0* as $D_a$ grows, making preservation of a nonzero cross-modal inner product *impossible* under the stated construction. This invalidates the paper’s stated Theorem 1 (cross-modal inner-product preservation under block-diagonal projection) and any corollaries that rely on it (e.g., PMME’s “geometry-preserving” property).

2. **Evaluation Gaps**:
    1. *Lack of Direct Baseline Comparison:* The evaluation is missing a direct comparison against a key alignment-based baseline, Mask Grounding, within the same experimental setting. Since AML is presented as a plug-and-play, alignment-aware strategy similar to Mask Grounding, its claimed universality should be validated by outperforming this baseline under identical, fully trained conditions. Table 2, however, only reports CARIS/DETRIS add-ons and omits Mask Grounding, leaving this critical parity untested. The table should have directly compared Mask Grounding on CARIS/DETRIS with AML on CARIS/DETRIS.
    2. *Omission of MagNet mIoU Score:* Table 1 omits the mIoU score for MagNet, an important alignment baseline.

3. **Training Efficiency:** Training and memory costs from implementing AML should be put in the main paper, not the appendix.

4. **Generalization / Real-world robustness:** The paper's visualizations are limited to the heavily optimized RefCOCO datasets, making it difficult to assess performance on unconstrained "in-the-wild" images. Including qualitative results on such images, mirroring the style of Figure 4(a), is essential to demonstrate the model's real-world generalization

5. **Reproducibility:** Since code is not given, adding some crucial pseudo code in the paper will be tremendously helpful for the broader audience to reproduce this work.

6. **Many Writing & Formatting Issues**:
    1. *References style:* inconsistent reference format; inconsistent capitalization (e.g., an author name in ALL CAPS on line 594); duplicate/inconsistent entries (e.g., CRIS 2022a/2022b). Standardize to ICLR format.
    2. *Typos and formatting errors:* e.g. casing of softmax/Softmax is sometimes small (equation 16) and sometimes large (equation 6); no spacing in Theorem1 and Theorem2; PPME in line 1106.
    3. *Unconventional figure naming:* Figure a, Figure b, Figure c etc.
    4. *Readability of text in figures:* Some texts in Figure 1, Figure a and Figure b are too small.

**Questions:**

Refer to weaknesses.

---

> ### Author Response · Authors · 2025-11-23
>
> Thank you for your detailed comments and helpful feedback. Below, we provide our responses to each of your comments.
>
> ## W1 On theory clarification
> Thank you for the careful reading of our theoretical analysis. After re-checking the derivation, we have identified a few notational inaccuracies in Appendix F.2, Step 3. These issues do not affect the correctness of our main result. Below we provide a precise correction, followed by clarification on the reviewer’s concerns regarding Gaussian random projections.
>
> According to Theorem 2 (see Appendix F.3), we know that:
>
> $$(1-\varepsilon)\|z - z'\|^2
> \le \|\widetilde{W}z - \widetilde{W}z'\|^2
> \le (1+\varepsilon)\|z - z'\|^2 .$$
>
> Then:
>
> $$
> \bigl| \langle \widetilde{W}z, \widetilde{W}z' \rangle - \langle z,  z' \rangle \bigr|
> \le
> \frac{1}{2}\bigl| \|\widetilde{W}z - \widetilde{W}z'\|^2 - \|z - z'\|^2 \bigr|
> \le
> \frac{1}{2} \varepsilon \cdot \|z - z'\|^2 .
> $$
>
> Given $\|z - z'\| \leq 2$, thus，
> $$
> \bigl| \langle \widetilde{W}z, \widetilde{W}z' \rangle - \langle z, z' \rangle \bigr|
> \le
> \frac{1}{2} \varepsilon \cdot \|z - z'\|^2 \le 2\varepsilon
> $$
> Thus the geometric distortion between samples before and after projection is bounded by $2\varepsilon$, which is the foundation of our approximate modality-lifting guarantee. As for Line 883- Line 889, that should be deleted and and the content has been updated. We thanks you for pointing this.
>
> **On the your concern:** $\mathbb{E}[\langle Wv, Wu \rangle] = 0, \operatorname{Var}(\langle Wv, Wu \rangle) = \frac{1}{D_a}$ .
> We provide three clarifications:
>
> 1. Theorem 1 concerns the joint sample space, not cross-modal inner products. Theorem 1 examines the stability of the concatenated vector  $z = [u,v],$ and establishes that pairwise distances  $\|z_i - z_j\|$ are preserved after block-diagonal random projection. Importantly, Theorem 1 **does not require**
> $\mathbb{E}\big[\langle W_i v, W_t u \rangle\big] \neq 0.$ Instead, it guarantees **geometry preservation** in the *joint sample space*, ensuring that the optimization remains valid even after projection.
> This is exactly the sense in which “geometry-preserving” applies in our setting.
>
> 2. Although $\mathbb{E}[\langle Wv, Wu\rangle] = 0$, PMME does not rely on absolute inner-product magnitude You are correct about for two independent Gaussian random projections, $\mathbb{E}[\langle Wv, Wu \rangle] = 0, \operatorname{Var}(\langle Wv, Wu \rangle) = \frac{1}{D_a}$, However, this does **not** harm PMME, because PMME computes: $\text{PMME} = \max_k \text{SoftMax}\big(V' {T'}^\top\big),$ (Eq.5-Eq.6 in Sec.3.2)
> i.e., a **row-wise SoftMax followed by Max** . Thus PMME uses **relative discriminability**, not absolute dot-product magnitude.
> Even when all raw projected inner products are small, their *relative differences* remain detectable, enabling reliable token–patch matching. In other words, PMME only requires *contrast*, not large values.
>
> 3. No-free Launch. Your insight reveals a meaningful trade-off, since $W_v$ and $W_u$ are independent Gaussian projections,
> $\langle Wv, Wu \rangle \to 0 \quad \text{as } D_a \to \infty.$ This produces a fundamental trade-off:
> - Larger $D_a$ → **better JL distance preservation** (Theorem 1).
> - Larger $D_a$ → **smaller cross-modal variance**, reducing contrast.
> This “no-free-lunch” behavior motivates analyzing performance under different projection dimensions.
>
> | Dim ($D_a$) | oIoU | mIoU | Pr@50 | Pr@90 | JL Error Bound ($\varepsilon$) |
> |:-------------:|:----:|:----:|:-----:|:-----:|:-------------------------------:|
> | 1024          | 66.82 | 70.83 | 80.15 | 36.27 | 0.2955 |
> | 2048          | 67.37 | 71.33 | 80.82 | 36.35 | 0.2088 |
> | 4096          | 67.08 | 71.21 | 80.85 | 36.22 | 0.1478 |
>
> Increasing further to 4096 yields diminishing returns, aligning with the trade-off.
>
> ## W2.1 On the missing comparison with Mask Grounding
> We agree that Mask Grounding is a relevant alignment-based method. However, Mask Grounding does not open-source its training code, and key training parameters, making reproducibility hard. Instead, we (i) compare against MagNet in its original setting using the authors’ official numbers, and (ii) demonstrate the plug-and-play universality of AML by integrating it into three diverse RIS backbones (CARIS, DETRIS, ReLA) without modifying their architectures or loss functions. Conceptually, Mask Grounding and AML act on different sides of the pipeline:
> - Mask Grounding improves text features by predicting masked tokens with visual + mask cues;
> - AML operates purely on the vision side, using per-epoch patch–token similarity to down-weight low-alignment pixels in the segmentation loss, with no extra modules, no extra losses, and no inference-time overhead.
> These are therefore complementary rather than equivalent mechanisms, and AML could in principle be combined with Mask Grounding in future work.

---

> ### Author Response · Authors · 2025-11-23
>
> ## W2.2 On the omission of MagNet mIoU in Table 1
>
> The absence of MagNet’s mIoU was not intentional. In the 50-epoch single-dataset setting that we adopted, the MagNet paper reports oIoU only and does not provide mIoU under the same protocol. Following your suggestion, we have tested MagNet using the official testing scripts and weights provided by the authors and updated the results in line 392 of the manuscript.
>
> ## W3 & W4 & W5 Training Efficiency, Generalization / Real-world robustness, Reproducibility
> We appreciate your suggestions regarding the presentation of our manuscript. In response to your comments on W3, W4, and W5, we have added the modifications in lines 510-518, 1332-1335 and 324-360 of the manuscript.
>
> ## W6 Many Writing & Formatting Issues
> We thank the reviewer for pointing out the inconsistency in writing and formatting. In the updated version, we have standardized the references style, corrected the formatting of equations, appendix tables, and figure labels, and adjusted or removed the small text in previous Figures 1, Figure a, and Figure b.

---

### Official Review · Reviewer_Vg5H · 2025-10-29

**Soundness:** 2
**Presentation:** 2
**Contribution:** 2
**Rating:** 4
**Confidence:** 4

**Summary:**

- This work proposes Alignment-Aware Masked Learning (AML), a training strategy that quantifies region–referent alignment (PMME) and filters out unreliable pixels during optimization (AFM), which is validated to improve RIS performance.

**Strengths:**

- This work proposes Alignment-Aware Masked Learning (AML), a training strategy that quantifies region–referent alignment (PMME) and filters out unreliable pixels during optimization (AFM), which is validated to improve RIS performance.

- The writing overall is good and it is easy for readers to understand the proposed framework.

- The experiments analysis is detailed for readers  to realize the benefits of AML.

**Weaknesses:**

- Motivation clarification: the motivation is not well clarified from figure-1.  I suppose the author's motivation is that a number of regions (especially background regions) dominate the training loss.

- Method contributions
  - Based on the above motivation, I am more inclined to believe that this work is actually an implementation of curriculum learning in the RIS task. It is also be validated from the efficiency of early-training stage. In view of the originality, it decreases the contributions of this work.
  - The work claims that the stage-I is forward-only. I am curious how the similarity of these raw visual-language features reflects the degree of alignment.
  - The whole performance improvement is weak especially on the more strong models, which makes the work seem less significant at present community.

- Some writings are confusing.
  - The explanation about $B^h$ and  $B^w$ (line-267).
  - For the `early-stage efficiency', to my knowledge, this is not a common term and it deserves a specific explanation.

- Extra experiments for validation
  - The author can verify the results of using different models at different stages (e.g., utilize ReLA for stage-1, CARIS for stage-2), which may bring some new observations.

**Questions:**

Refer to the Weaknesses.

---

> ### Author Response · Authors · 2025-11-23
>
> We sincerely appreciate your careful review and insightful suggestions. Below, we provide our responses to each of your comments.
> ## W1 Motivation and Figure 1
>
> Your understanding is generally correct, but in the context of the RIS task, the background should be interpreted as referring to irrelevant regions. This not only includes typical background areas (e.g., sky, rooms, grass) but also objects that are visually similar yet irrelevant to the referring expression. This distinction means that our optimization focuses on refining the decision boundaries between semantically similar candidates, rather than large, information-sparse, irrelevant regions. As shown in Figure 1 (a–b), the CARIS baseline often struggles to clearly differentiate between highly similar objects (e.g., multiple giraffes, stacked vegetables), even when the global context is correct, leading to uncertain or leaking predicted boundaries. The improvements in both baseline performance and robustness, as demonstrated in Figure 1 (c–d), empirically support this motivation.
>
> ## W2.1 Relation to Curriculum Learning and Contribution
> We agree that AML shares similarities with curriculum learning in improving early-stage optimization, but our mechanism differs in key ways:
>
> - We do not define a fixed difficulty schedule over samples.
> - Masking is computed dynamically per image and epoch based on the cross-modal similarity map from PMME, with no manual "easy → hard" transition.
> - The core principle is alignment-aware denoising: weakly aligned pixels are filtered out, while strongly aligned referent and context regions are retained.
>
> In early training, when cross-modal alignment is weak, we leverage the unimodal structure of pretrained Swin-B and BERT, along with our JL random projection (Sec. 3.2, Thm. 1), to generate a noisy yet informative alignment signal. As training progresses with referent supervision, the model's alignment improves, and the masking rule evolves to suppress irrelevant regions and focus on ambiguous, similar objects. Thus, AML is not an RIS-specific curriculum, but a generic alignment-aware masking approach based on the model’s current feature space.
>
> ## W2.2 How Stage-I Similarity Reflects Alignment
> Although stage-I is forward-only, the similarity used by PMME is not arbitrary:
> - The visual and textual encoders (Swin-B and BERT) are individually pretrained, so both modalities already carry rich semantic structure.
> - PMME first ℓ₂-normalizes features and maps them into a shared Da-dimensional space via fixed Gaussian projections. As proven in Thm. 1, this Johnson–Lindenstrauss mapping approximately preserves all cross-modal inner products with high probability, ensuring that similarity in the projected space is a faithful proxy of similarity in the original feature spaces.
>
> - We then take, for each patch, the maximum similarity over tokens, yielding a fine-grained “how well can this patch match any word” score.
> Empirically, Table 7 shows that random (fixed) projection combined with PMME already yields substantial oIoU gains over both the baseline and a learnable projection variant at the first epoch, indicating that PMME captures meaningful alignment signals even before joint fine-tuning. The alignment statistics and similarity distributions in Appendix G.4 further confirm that our similarity scores become sharper and more concentrated around truly text-relevant regions over training.

---

> > ### Comment · Reviewer_Vg5H · 2025-11-24
> > **Why not use the aligned VLM models for similarity measure.**
> >
> > Thanks for your response.
> > Some concerns about PMME  still exists. I don't quite grasp the practical significance and feasibility of the PMME.
> > - If PMME can connect the feature spaces of two separately pre-trained unimodal models, could it then replace multimodal feature alignment models like CLIP and SLIP in multimodal tasks?
> >
> > - If not, why not directly use pre-aligned multimodal models instead of adopting PMME?

---

> > > ### Author Response · Authors · 2025-11-24
> > >
> > > Thank you for your thoughtful feedback. We are eager to engage in a discussion on this topic. In fact, it depends on what role CLIP plays in tasks like Referring Image Segmentation (RIS). We address PMME's positive impact from two perspectives:
> > >
> > > ## 1.Using CLIP to Generate Masks
> > > We believe this explains the main difference in our viewpoints. The alignment that the model learns may not exactly match the alignment predefined by humans. While using CLIP to guide the mask towards semantically irrelevant regions may provide some improvements, it does not perform as well as our PMME approach.
> > >
> > > As shown in the table below, the performance of pre-trained model with AML and PMME trained from a stable alignment stage (from epoch 5) is comparable but still does not outperform the full use of PMME.
> > >
> > > | Method                        | oIoU  | mIoU  |
> > > |-------------------------------|-----|-------|
> > > | **Pre-trained Model with AML** |  67.04 | 70.89 |
> > > | **AML (from 5 epochs)**        | 67.08 (+0.04) | 70.84 (-0.05) |
> > > | **AML (at all epochs)**        | 67.37 (+0.33) | 71.33 (+0.44) |
> > >
> > > This demonstrates that generating masks based on model-specific alignment (as PMME does) yields better results than relying on externally defined alignment, such as CLIP. This intuitively points to a simple concept: the model learns better at understanding and structuring information within its "intrinsic semantic space." Therefore, masks generated based on the model’s internal alignment rules better match the training dynamics than forcing the use of external, human-defined alignments (like CLIP).
> > >
> > > ## 2.Using CLIP as a Pretrained Backbone
> > > We do not view PMME as a general image-text embedding model. However, in the RIS domain, using a general-purpose model like CLIP as the initial backbone does not yield ideal results.
> > >
> > > For example, as shown in table below, methods such as CRIS[1], CM-MaskSD[2], and RISCLIP[3] demonstrate that, even with CLIP as the backbone, the advantages of "semantic alignment" and shared embedding space do not directly translate into top-tier performance in RIS tasks.
> > >
> > > | Method / Dataset   | Text Backbone | Visual Backbone   | RefCOCO val | RefCOCO testA | RefCOCO testB | RefCOCO+ val | RefCOCO+ testA | RefCOCO+ testB | RefCOCOg val | RefCOCOg test |
> > > |--------------------|---------------|-------------------|-------------|---------------|---------------|--------------|----------------|----------------|--------------|---------------|
> > > | CRIS               | CLIP          | CLIP-RN101        | 70.47       | 73.18         | 66.10         | 62.27        | 68.08          | 53.68          | 59.87        | 60.36         |
> > > | CM-MaskSD          | CLIP          | CLIP-ViT-Base     | 72.18       | 75.21         | 67.91         | 64.47        | 69.29          | 56.55          | 62.67        | 62.69         |
> > > | RISCLIP-B          | CLIP          | CLIP-ViT-Base     | 73.57       | 76.46         | 69.76         | 65.53        | 70.61          | 55.49          | 64.10        | 65.09         |
> > > | CARIS              | BERT          | Swin-B            | 74.65       | 77.83         | 71.70         | 65.54        | 71.86          | 57.97          | 65.15        | 65.00         |
> > > | CARIS + AML        | BERT          | Swin-B            | 75.45       | 78.33         | 72.12         | 67.37        | 73.19          | 59.51          | 65.67        | 66.78         |
> > >
> > >
> > > This could be due to the finer feature resolution and interaction mechanisms required for spatial segmentation and region recognition tasks in RIS. In contrast, state-of-the-art RIS methods tend to use more specialized unimodal pretrained models, such as DINO/DINOv2, ViT, and Swin for vision, paired with text models like BERT and CLIP. PMME integrates well with these methods, suggesting that it may have deeper practical value in this domain.
> > >
> > > ---
> > >
> > > References:
> > >
> > > [1] Wang, Z., Lu, Y., Li, Q., Tao, X., Guo, Y., Gong, M., & Liu, T. (2022). CRIS: Clip-driven Referring Image Segmentation. In *IEEE Conference on Computer Vision and Pattern Recognition (CVPR)*.
> > >
> > > [2] Wang, W., He, X., Zhang, Y., Guo, L., Shen, J., Li, J., & Liu, J. (2024). CM-MaskSD: Cross-modality Masked Self-Distillation for Referring Image Segmentation. *IEEE Transactions on Multimedia, 26*, 6906–6916.
> > >
> > > [3] Kim, J., Lee, H., & Park, Y. (2024). Extending CLIP's Image-Text Alignment to Referring Image Segmentation. In *North American Chapter of the Association for Computational Linguistics (NAACL)*.

---

> ### Author Response · Authors · 2025-11-23
>
> ## W2.3 Magnitude and Significance of Improvements
>
> We interpret performance differently. While achieving SoTA results is important, and we’ve achieved that, it’s also essential to push the community forward by exploring more advanced backbones and tackling more challenging settings:
>
> - On DETRIS, a parameter-efficient DINOv2–CLIP based model, AML still improves oIoU on all RefCOCO+ splits (Table 2).
>
> -  AML also brings substantial robustness gains: when training on RefCOCO+ and testing under seven visual perturbations on RefCOCO/RefCOCOg, AML improves average mIoU by +3.50 and +2.34, respectively (Table 15 in Appendix G.5).
>
> - On CARIS , AML yields consistent improvements across all 8 splits: e.g., +2.00 / +1.92 mIoU on RefCOCO+ val/testB and +1.83 / +1.54 oIoU (Table 1).
>
> ## W3.1 & 3.2 Some Writings are Confusing
>
> We have modified thess explanations at lines 267~268 and 411 at modified version.
>
> ## W4 Using Different Models at Different Stages
>
> We believe that the reviewer is suggesting the use of a pre-trained model that already exhibits strong cross-modal alignment. To address this, we conducted experiments as shown below:
>
> | Method                        | P@0.5  | P@0.7  | P@0.9  | oIoU  | mIoU  |
> |-------------------------------|--------|--------|--------|-------|-------|
> | **Pre-trained Model with AML** | 80.30  | 73.59  | 36.31  | 67.04 | 70.89 |
> | **AML (from 5 epochs)**        | 80.22  | 73.67  | 36.00  | 67.08 | 70.84 |
> | **AML (at all epochs)**        | 80.69  | 74.17  | 36.76  | 67.37 | 71.33 |
>
> We found that the performance of **Pre-trained Model with AML** is comparable to **AML (from 5 epochs)**. This suggests that using a model with better initial alignment results in performance improvements. However, for our method, **early-stage application of AML** remains important. Our approach benefits significantly from the early refinement of alignment, which is why starting from earlier epochs leads to better performance than later-stage application, as shown in the results.

---

### Official Review · Reviewer_P4qH · 2025-10-30

**Soundness:** 4
**Presentation:** 4
**Contribution:** 3
**Rating:** 8
**Confidence:** 5

**Summary:**

This paper presents an alignment-aware masked learning method for referring image segmentation. In particular, each sample is masked out by discarding pixels below an adaptive similarity threshold. The similarity map between visual and textual features is quantified by region-referent alignment. The framework is then trained after the first masking step. The above two steps are conducted in an interleaved fashion. In addition, the region-referent alignment is implemented via a PatchMax Matching Evaluation strategy on randomly projected visual and textual features. Experimental results validated the effectiveness of the proposed method.

**Strengths:**

(1) The motivation is well presented of using the proposed alignment-aware masked learning approach for referring image segmentation.

(2) The explanations and illustrations are mostly clear and intuitive of the PatchMax Matching Evaluation, the alignment-aware filtering mask and the training strategy.

**Weaknesses:**

(1) The approach of using a previous-step inference for mask prediction and guide the current learning may face convergence issue. In fact, the initial state of mask is largely incorrect and can result in unexpected learning curves. There is no discussion on this issue.

(2) On the fairness of experimental comparison, since CARIS+AML uses 17.2% more training time than CARIS (according to Appendix G.2), the performance gain in Table 1 is also possibly coming from longer training. There is no ablation study and discussion on this issue.

**Questions:**

No.

---

> ### Author Response · Authors · 2025-11-23
>
> We appreciate your understanding of our work and the positive feedback. Below, we provide further responses to your concerns.
>
> ## W1. Convergence Issue with Pre-step Inference for Mask Prediction
>
> We acknowledge the concern that initial mask may select at semantic relevant regions.. However, we argue that the early masks reflect an inherent initial alignment tendency based on the feature space constructed by the model at its current stage, rather than incorrect predictions. These masks guide the model's focus toward regions with easier alignments, which are essential for stable early-stage convergence.
>
> At the same time, as shown in the table below, the proportion of coverage on the target during the entire training phase is quite small and steadily decreases. This demonstrates that our masking strategy becomes increasingly precise as training progresses.
>
> | **$\tau$ / Epoch** | **0**    | **1**    | **2**    | **3**    | **4**    | **5**    | **6**    | **7**    | **8**    | **9**    | **10**   |
> |-------------------|----------|----------|----------|----------|----------|----------|----------|----------|----------|----------|----------|
> | **0.20**          | 0.0021   | 0.0007   | 0.0004   | 0.0005   | 0.0005   | 0.0005   | 0.0005   | 0.0005   | 0.0004   | 0.0005   | 0.0004   |
> | **0.30**          | 0.0097   | 0.0057   | 0.0058   | 0.0053   | 0.0052   | 0.0051   | 0.0049   | 0.0044   | 0.0046   | 0.0043   | 0.0043   |
> | **0.40**          | 0.0259   | 0.0178   | 0.0157   | 0.0139   | 0.0135   | 0.0129   | 0.0130   | 0.0125   | 0.0121   | 0.0122   | 0.0120   |
>
> To demonstrate the effectiveness of our method in the early stages, we conducted experiments where AML was applied starting from different training epochs.
>
> | Methods                     | P@0.5  | P@0.7  | P@0.9  | oIoU  | mIoU  |
> |-----------------------------|--------|--------|--------|-------|-------|
> | **CARIS**                   | 78.85  | 71.76  | 33.48  | 65.54 | 69.33 |
> | **AML (from 20 epochs)**    | 79.81  | 73.02  | 35.68  | 66.81 | 70.45 |
> | **AML (from 5 epochs)**     | 80.22  | 73.67  | 36.00  | 67.08 | 70.84 |
> | **AML (at all epochs)**     | 80.69  | 74.17  | 36.76  | 67.37 | 71.33 |
>
> As shown in the table, we observe that AML (all epochs) outperforms AML (from 5 epochs), which in turn performs better than AML (from 20 epochs), particularly at higher precision thresholds (e.g., precision@0.7 and precision@0.9). This highlights that applying AML throughout all epochs yields the best performance, with earlier application leading to improved results compared to later stages. The superior performance in early epochs further supports that early use of AML is not only beneficial but necessary for achieving better alignment and segmentation precision.
>
> The optimization process, influenced by these initial masks, effectively prioritizes well-aligned features while filtering out difficult-to-align regions, thereby improving early optimization efficiency. The attention mechanism further refines this alignment during later stages, ensuring that the model adjusts its focus as it progresses through training.
>
> ---
>
> ## W2. Fairness of Experimental Comparisons Due to Increased Training Time
>
> While CARIS + AML introduces a **17.2% increase in training time** due to an additional forward pass (see **Appendix G.2**), the total number of optimization steps remains unchanged. The extra computational cost does not result in additional optimization iterations.
>
> ---
>
>
> | Method                   | Memory (MB)        | Time/Epoch (min)  | 10 epochs       | 30 epochs       | 50 epochs       |
> |--------------------------|--------------------|-------------------|-----------------|-----------------|-----------------|
> | CARIS                    | 35,168             | 30.16             | 60.2 / 62.3     | 64.7 / 68.3     | **65.5 / 69.3**     |
> | +AML (same time)         | 36,902 (+4.9%)| 35.35 (+17.2%)| 61.8 / 64.2     | **65.3 / 69.5**     | 66.7 / 70.8     |
> | +AML (same steps)        | 36,902 (+4.9%) | 35.35 (+17.2%) | 63.8 / 65.4 | 66.1 / 70.1 | 67.4 / 71.3 |
>
> ---
>
> As shown in table,  under the same wall-clock time (8/26/42 epochs for CARIS+AML vs. 10/30/50 for CARIS), AML consistently outperforms the baseline (e.g., **66.7/70.8** vs. **65.5/69.3** at the 50-epoch time budget), and already reaches the 50-epoch CARIS performance using roughly the 30-epoch time budget (**65.3/69.5** vs. **65.5/69.3**). In the equal-training-step setting (10/30/50 epochs for both), CARIS+AML also yields consistent gains (up to **67.4/71.3** vs. **65.5/69.3** at 50 epochs), confirming that the improvements come from the **alignment-aware masking strategy** rather than from additional optimization steps. We would like to thank the reviewers for their valuable feedback. In response to their suggestion, we have incorporated the overhead analysis into the main body of the text.

---

### Official Review · Reviewer_pnob · 2025-11-02

**Soundness:** 3
**Presentation:** 3
**Contribution:** 3
**Rating:** 6
**Confidence:** 4

**Summary:**

This work proposes a novel framework called AMLRIS, which aims to improve the performance of Referring Image Segmentation (RIS). The framework first introduces a Johnson-Lindenstrauss random projection to measure the similarity between image representations and token features. Then, the image pixels with low similarity are filtered out from the training process to stabilize the training and improve performance. Experimental results demonstrate that AMLRIS achieves superior performance compared to standalone training, and the ablation experiments show the effectiveness of the proposed modules.

**Strengths:**

• The projection design provides a novel approach for measuring similarity between representations of different modalities. This method can be extended to more tasks.
• Experiments demonstrate the effectiveness of the proposed structure, achieving competitive results across multiple downstream datasets.
• The proposed structure does not significantly increase training overhead while maintaining inference time.
• The proposed idea is interesting and generally well-motivated, and the experimental evaluation is relatively thorough

**Weaknesses:**

My primary concern is the method’s sensitivity to small or low-contrast objects. The AML
framework relies on PMME to generate alignment-based masks by identifying high-confidence visual patches. This mechanism inherently depends on the relative distribution of features within the image. As a result, small objects or objects with low visual saliency may produce low peak alignment scores and be incorrectly masked out during training. Consequently, the model’s performance may degrade on images where the target occupies a very small region or is visually subtle.
• In Figure 2, the masked pixels appear to be almost exclusively background regions. It is unclear
whether masking such areas truly helps the model focus on the target objects. In my view, it would be more important to mask regions corresponding to potentially confusing objects rather than background. This raises some concerns regarding the effectiveness of PMME in guiding the model’s attention.
• The mechanisms underlying some of the core components of the model remain unclear. I would be willing to consider a higher score if the authors provide clear explanations addressing my concerns.

**Questions:**

See Weakness above.

---

> ### Author Response · Authors · 2025-11-23
>
> Thanks for your positive feedback, here are our response to your remaining concerns.
>
> ## W1. Performance Degradation on Small or Low-Contrast Objects
>
> Our masking strategy is designed to focus on masking regions that the model deems to have low relevance to the textual description. For small or visually insignificant objects, this mechanism selectively identifies which areas to mask. If the features of a small object are strongly aligned with the description, our approach retains and emphasizes those features. On the other hand, if an object is visually inconspicuous and lacks contextual aggregation in that region, it is likely to contain less discriminative information. In such cases, we reduce focus on these regions and shift attention towards more relevant, discriminative features within the target that offer higher optimization potential.
>
> ### Performance Comparison on Lowlight and Haze Transformations
>
> | Transformation  | mIoU / oIoU / Pr@0.5 | mIoU / oIoU / Pr@0.5 | Improvement |
> |-----------------|-------------|---------------------|-----------------|
> | **Haze**        | 59.33 / 57.85 / 70.18 | 62.54 / 60.63 / 73.32 | +3.21 / +2.78 / +3.14 |
> | **Lowlight**    | 58.32 / 56.81 / 69.07 | 61.63 / 59.81 / 72.46 | +3.31 / +3.00 / +3.39 |
>
> Based on the results for low-contrast scenes like Lowlight and Haze, our method (AML) consistently shows improvements in all metrics, demonstrating its effectiveness in challenging scenarios where visibility or feature distinction is limited.
>
>
>
> ## W2. Effectiveness of PMME in Guiding Model Attention
>
> We appreciate the reviewer’s feedback on the effectiveness of PMME in guiding model attention. In Figure 2, the masked pixels primarily represent background regions, raising concerns about whether masking these areas improves focus on the target objects. While it may not seem intuitive, our approach emphasizes distinguishing relevant features rather than eliminating information.
>
> The core goal of PMME is to guide the model to focus on regions most relevant to the reference expression. We believe that, rather than masking confusing or overlapping regions entirely, focusing on them can enhance the model's discriminative power. By encouraging the model to attend to prominent features that clearly differentiate the referent from other objects, our method refines the model’s ability to discern key details.
>
> Additionally, defining the "right" regions to mask is challenging because critical discriminative information often lies in ambiguous or surrounding areas. For example, in the description "the giraffe closest to the man," the giraffe is the referent, but the surrounding human context is essential for disambiguation. Masking out these areas would remove important contextual information, hindering optimization.
>
> Instead of masking potentially confusing regions, we retain important visual information while reducing focus on irrelevant areas. This strategy enables the model to concentrate on key discriminative features while preserving the context of the referent, without relying on predefined regions. It allows the model to dynamically learn which areas to prioritize, improving both efficiency and flexibility during training.
>
> We hope these clarifications address your concerns, and we appreciate the opportunity to explain the underlying mechanisms of our approach.

---

### Author Response · Authors · 2025-11-23
**Unified Summary of Common Concerns**

We thank the reviewers for their thoughtful feedback and first clarify the common concerns on AML’s motivation, effectiveness mechanism, and training overhead.


### **Clarification of the Motivation**

We appreciate the reviewer’s comments. Figure 1 demonstrates the core challenge: CARIS struggles with semantically similar or overlapping objects (e.g., multiple giraffes or stacked vegetables), leading to uncertain or overlapping boundaries.

The issue arises from **full-pixel supervision**, where the model fails to focus on fine-grained regions related to the text and is distracted by background areas with weak alignment. This results in the model focusing on one-to-one correspondences, affecting the precision of segmentation boundaries and textual understanding.

Our AML method addresses this by **dynamically adjusting training masks** to prioritize regions aligned with the text, especially in early stages. This improves segmentation precision, as shown in **Appendix D** and **Figure 4**, while **Appendix G.4** confirms the progressive improvement in alignment. We also tested our method on various visual scenarios and images beyond RefCOCO (e.g., **Figure 8**), showing consistent and robust results.

---

### **Clarification of the Effectiveness Mechanism**

Our AML method can be compared to simple curriculum learning, but with a fundamental difference: while curriculum learning typically follows a fixed "easy-to-hard" strategy, our approach is based on a dynamic, adaptive optimization process. Instead of manually creating "easy" and "confusing" samples through masking, we dynamically filter out visual information with excessive semantic divergence based on the model's current understanding of modality alignment. To demonstrate the effectiveness of our method in the early stages, we conducted experiments where AML was applied starting from different training epochs.

### Performance Comparison

| Methods                     | P@0.5  | P@0.7  | P@0.9  | oIoU  | mIoU  |
|-----------------------------|--------|--------|--------|-------|-------|
| **CARIS**                   | 78.85  | 71.76  | 33.48  | 65.54 | 69.33 |
| **AML (from 20 epochs)**    | 79.81  | 73.02  | 35.68  | 66.81 | 70.45 |
| **AML (from 5 epochs)**     | 80.22  | 73.67  | 36.00  | 67.08 | 70.84 |
| **AML (at all epochs)**     | 80.69  | 74.17  | 36.76  | 67.37 | 71.33 |

As shown in the table, we observe that AML (all epochs) outperforms AML (from 5 epochs), which in turn performs better than AML (from 20 epochs), particularly at higher precision thresholds (e.g., precision@0.7 and precision@0.9). This highlights that applying AML throughout all epochs yields the best performance, with earlier application leading to improved results compared to later stages.

In our analysis, although the model's overall alignment is low in the early stages, it uses PMME similarity to generate masks that filter out poorly aligned regions and reinforce visual features that better align with the textual representation. As training progresses, the dynamically generated masks increasingly guide optimization toward the true semantic boundaries, creating a virtuous cycle of optimization, feedback, and focus, which ultimately leads to finer segmentation.

Therefore, our method is a dynamic masking mechanism that adapts to the model’s cognitive state, continuously refining the alignment of relevant regions to achieve precise segmentation. The superior performance in early epochs further supports that early use of AML is not only beneficial but necessary for achieving better alignment and segmentation precision.

---

### Author Response · Authors · 2025-11-23

### **Clarification of the Overhead**

While CARIS + AML introduces a **17.2% increase in training time** due to an additional forward pass (see **Appendix G.2**), the total number of optimization steps remains unchanged. The extra computational cost does not result in additional optimization iterations.

---


| Method                   | Memory (MB)        | Time/Epoch (min)  | 10 epochs       | 30 epochs       | 50 epochs       |
|--------------------------|--------------------|-------------------|-----------------|-----------------|-----------------|
| CARIS                    | 35,168             | 30.16             | 60.2 / 62.3     | 64.7 / 68.3     | **65.5 / 69.3**     |
| +AML (same time)         | 36,902 (+4.9%)| 35.35 (+17.2%)| 61.8 / 64.2     | **65.3 / 69.5**     | 66.7 / 70.8     |
| +AML (same steps)        | 36,902 (+4.9%) | 35.35 (+17.2%) | 63.8 / 65.4 | 66.1 / 70.1 | 67.4 / 71.3 |

---

As shown in table,  under the same wall-clock time (8/26/42 epochs for CARIS+AML vs. 10/30/50 for CARIS), AML consistently outperforms the baseline (e.g., **66.7/70.8** vs. **65.5/69.3** at the 50-epoch time budget), and already reaches the 50-epoch CARIS performance using roughly the 30-epoch time budget (**65.3/69.5** vs. **65.5/69.3**). In the equal-training-step setting (10/30/50 epochs for both), CARIS+AML also yields consistent gains (up to **67.4/71.3** vs. **65.5/69.3** at 50 epochs), confirming that the improvements come from the **alignment-aware masking strategy** rather than from additional optimization steps. We would like to thank the reviewers for their valuable feedback. In response to their suggestion, we have incorporated the overhead analysis into the main paper.

---

### Meta-Review · Area_Chair_Nfo1 · 2026-01-04

**Summary:**

AMLRIS proposes an alignment-aware masked learning strategy (PMME + AFM) for RIS that is plug-and-play and shows consistent gains on RefCOCO/+/g plus robustness tests. Post-rebuttal, the main remaining risk is soundness/interpretability of PMME and the theoretical framing, along with whether the paper sufficiently positions itself against closely related “alignment-guided masking/grounding” baselines. Reporting GRES results on gRefCOCO is suggested. Overall, the AC leans Accept.

**Reviewer Concerns:**

Addressed: fairness about extra training time (same-time/same-steps comparisons), convergence/when AML is applied, added/filled missing metrics (e.g., MagNet mIoU), added training/memory overhead into main text, and cleaned many writing/format issues.

Still outstanding: lingering skepticism on PMME’s practical meaning vs using pre-aligned VLMs (e.g., CLIP), and partial concern that the theory section needed correction/clarification (even if authors revised it); plus missing direct apples-to-apples comparison with some key alignment-style baselines remains a potential gap.

**Reviewer Scores:**

- P4qH (8): likely 8 → 8 (core concerns addressed, already confident).
- pnob (6): likely 6 → 6 (questions answered).
- Vg5H (4): likely 4 → 4/5 (PMME feasibility concern may persist).
- X2d4 (4): likely 4 → 5 (theory + missing-metric + presentation fixes help, but baseline/theory skepticism may remain).

---

### Decision · Program_Chairs · 2026-01-26

Accept (Poster)